# Glutamylation of centrosomes ensures their function by recruiting microtubule nucleation factors

Shi-Rong Hong[1,7], Yi-Chien Chuang [1,7], Wen-Ting Yang [1,7], Chiou-Shian Song[1,7], Hung-Wei Yeh[2], Bing-Huan Wu[1], I-Hsuan Lin[3], Po-Chun Chou[3], Shiau-Chi Chen[1], Lohitaksh Sharma [1], Jui-Chen Lu[1], Rou-Ying Li[4], Ya-Chu Chang[1], Kuan-Ju Liao[5], Hui-Chun Cheng [2], Won-Jing Wang[3], Lily Hui-Ching Wang[5,6] & Yu-Chun Lin [1,4✉]

## Abstract

**Centrosomes are tubulin-based organelles that undergo glutamylation, a post-translational modification that conjugates glutamic acid residues to tubulins. Although centrosomal glutamylation has been known for several decades, how this modification regulates centrosome structure and function remains unclear. To address this long-standing issue, we developed a method to spatiotemporally reduce centrosomal glutamylation by recruiting an engineered deglutamylase to centrosomes. We found that centrosome structure remains largely unaffected by centrosomal hypoglutamylation. Intriguingly, glutamylation physically recruits, via electrostatic forces, the NEDD1/CEP192/γ-tubulin complex to centrosomes, ensuring microtubule nucleation and proper trafficking of centriolar satellites. The consequent defect in centriolar satellite trafficking leads to reduced levels of the ciliogenesis factor Talpid3, suppressing ciliogenesis. Centrosome glutamylation also promotes proper mitotic spindle formation and mitosis. In summary, our study provides a new approach to spatiotemporally manipulate glutamylation at centrosomes, and offers novel insights into how centrosomes are organized and regulated by glutamylation.**

**Keywords** Centrosomes; Glutamylation; Microtubules; Primary Cilia; Centriolar satellites
**Subject Categories** Cell Adhesion, Polarity & Cytoskeleton; Post-translational Modifications & Proteolysis

## Introduction

Centrosomes are major microtubule-organizing centers, responsible for nucleating interphase microtubules and establishing mitotic spindles, and they serve as platforms for the extension of primary cilia and flagella (Werner et al, 2017). These functions are achieved via the recruitment of specific signaling molecules to centrosomes (Arquint et al, 2014). As a signaling hub, centrosomes participate in cell division, cell migration, and vesicular transport as well as genome stability (Etienne-Manneville and Hall, 2003). Defects in centrosomes give rise to several human disorders, including deafness, dysplasia, and retinitis as well as diseases such as obesity, diabetes, nephronophthisis, Parkinson disease, and various cancers (Badano et al, 2005).

Centrosomes are composed of two microtubule-based centrioles that are subject to glutamylation, a post-translational modification that adds glutamate chains onto the C-terminal tails of tubulins (Janke and Magiera, 2020). The extent of microtubule glutamylation in cells is modulated via the tight coordination of two different protein families, namely the tubulin tyrosine ligase-like (TTLL) family and cytosolic carboxypeptidase (CCP) family (Yang et al, 2021). The TTLL family initiates glutamylation and promotes elongation of the polyglutamate chain, while the CCP family proteins remove the chain. Through exquisite regulation of glutamylation level, the glutamate side chain acts as a tunable regulator of intrinsic microtubule stability and participates in extrinsic interactions with microtubule-associated proteins (MAPs) or motor proteins (Wloga et al, 2017; Lacroix et al, 2010; Valenstein et al, 2016; Suryavanshi et al, 2010; O'Hagan et al, 2017).

How glutamylation regulates modified microtubules has been studied in vitro. Purified microtubules chemically conjugated with glutamylated side chains enhance the motility of kinesin-1 motors, and microtubules modified by recombinant TTLL proteins modulate the binding of microtubule severing enzymes. These findings suggest that glutamylation may coordinate the behavior of MAPs and motors locally on modified microtubules (Sirajuddin et al, 2014; Valenstein et al, 2016). However, physiological evidence supporting the relevance of microtubule glutamylation at centrosomes is scarce, except for two early studies, in which microtubule glutamylation was masked by injecting GT335, an antibody against glutamylated microtubules, into cells; this perturbed centriolar integrity during mitosis (Bobinnec et al, 1998; Abal et al, 2005). The

[1]Institute of Molecular Medicine, National Tsing Hua University, Hsinchu 300044, Taiwan. [2]Institute of Bioinformatics and Structural Biology, National Tsing Hua University, Hsinchu 300044, Taiwan. [3]Institute of Biochemistry and Molecular Biology, National Yang Ming Chiao Tung University, Taipei 300093, Taiwan. [4]Department of Medical Science, National Tsing Hua University, Hsinchu 300044, Taiwan. [5]Institute of Molecular and Cellular Biology, National Tsing Hua University, Hsinchu 300044, Taiwan. [6]School of Medicine, National Tsing Hua University, Hsinchu, Taiwan. [7]These authors contributed equally: Shi-Rong Hong, Yi-Chien Chuang, Wen-Ting Yang, Chiou-Shian Song. ✉E-mail: ycl@life.nthu.edu.tw

main hurdle to understanding centrosomal glutamylation is the technical inability to precisely manipulate glutamylation density at centrosomes. Besides centrosomes, glutamylated microtubules are also enriched at ciliary axonemes in G0 cells and at mitotic spindles and intercellular bridges in dividing cells (Janke and Magiera, 2020). Conventional methods such as antibody masking of glutamylated microtubules and genetic manipulation of glutamylation enzymes perturb the entire cellular pool of glutamylated microtubules, thereby preventing specific functional analysis of centrosomal glutamylation (Yang et al, 2021). Moreover, these time-consuming methods cannot dissect the dynamic relationships among MAPs, glutamylation, and centrosomal properties. Therefore, to understand how centrosomes organize and regulate themselves via glutamylation, an approach that would enable precise manipulation of glutamylation at centrosomes would be highly desirable.

We previously established a method to spatiotemporally deplete tubulin glutamylation in ciliary axonemes via the rapid recruitment of an engineered deglutamylase from the cytosol to axonemes, which elucidated the specific roles of axonemal glutamylation (Yang et al, 2021; Hong et al, 2018). Building upon this method, here we recruited the engineered deglutamylase to centrosomes to rapidly reduce centrosomal glutamylation in living cells and simultaneously monitor the immediate effects on centrosome structural integrity and function.

## Results

### Rapid recruitment of an engineered deglutamylase to centrosomes for local glutamylation reduction

We first tested the possibility of manipulating centrosomal glutamylation by knocking down glutamylation enzymes. Among TTLL family members, TTLL5 was our target because it preferentially localizes to centrosomes and initiates glutamylation (Janke and Magiera, 2020; van Dijk et al, 2007; Yang et al, 2021). We knocked down *TTLL5* in NIH3T3 fibroblasts using two specific short-interfering RNAs (siRNAs) (Appendix Fig. S1A). *TTLL5* knockdown resulted in a subtle reduction in centrosomal glutamylation by ~16–20% (Appendix Fig. S1B,C). We assumed that other TTLLs may contribute to a compensatory mechanism of centrosomal glutamylation (Chawla et al, 2016). Moreover, many TTLLs are preferentially abundant in different tissues (Fullston et al, 2011; Grau et al, 2013; Ikegami et al, 2006; Wu et al, 2022), making it challenging to use TTLL knockdown to reduce centrosomal glutamylation in distinct systems. We therefore developed an alternative method to achieve a more efficient reduction of centrosomal glutamylation. The chemically inducible dimerization (CID) system has been widely used to manipulate cellular activities by recruiting proteins of interest (POIs) to target sites (DeRose et al, 2013; Fan et al, 2017). Rapamycin-triggered dimerization of two soluble proteins, namely FKBP (FK506 binding protein) and FRB (FKBP-rapamycin-binding domain), is one of the well-established CID systems (DeRose et al, 2013). Generally, FRB is fused with a specific targeting sequence to constrain its localization at a particular cellular site, whereas FKBP is tagged with a soluble POI that remains freely diffused in the cytosol under steady-state conditions. Rapamycin-induced FRB/FKBP

dimerization rapidly recruits POIs to target sites for localized reactions. To utilize CID to manipulate centrosomal glutamylation, we first explored centrosome-targeted proteins that preferentially localize to glutamylation-enriched regions of centrosomes. Several centrosomal proteins, including Centrin2, Chibby, CPAP, Kizuna, and the C-terminal portion of CEP170 (CEP170C), were tagged with green fluorescent protein (GFP) to visualize their distributions (Higgs and Peterson, 2005; Prosser and Morrison, 2015; Voronina et al, 2009; Tang et al, 2009; Oshimori et al, 2006). Among these candidates, GFP-CEP170C demonstrated the highest colocalization with glutamylated sites at centrosomes labeled by the antibody GT335 (Appendix Fig. S2A–D). Unlike the subdistal appendage-specific distribution of full-length CEP170 (Rodríguez-Real et al, 2023; Ma et al, 2022), CEP170C is evenly distributed at both mother and daughter centrioles (Appendix Fig. S2). Thus, GFP-CEP170C was chosen for subsequent experiments. Immunostaining results demonstrated that ectopic expression of a Cerulean3 fluorescent protein- and FRB-tagged CEP170C (hereafter Ce3-FRB-CEP170C) did not adversely affect centrosomal structure or function, as evidenced by the normal morphology of different centrosome subcompartments labeled by Centrin2 (centrioles), Pericentrin (pericentriolar matrix), CP110 (distal end of centrioles), ODF2 (subdistal appendages), CEP83 and CEP164 (distal appendages), and PCM1 (centriolar satellites) (Appendix Fig. S3A–O) as well as proper ciliogenesis (Appendix Fig. S3H,P) in Ce3-FRB-CEP170C-transfected cells. Collectively, these results confirmed that Ce3-FRB-CEP170C can target glutamylated sites at centrosomes without noticeable effects on centrosomes.

We next tested whether FKBP-tagged POIs could rapidly translocate from the cytosol to Ce3-FRB-CEP170C-labeled centrosomes upon rapamycin treatment (Fig. 1A). For this purpose, COS7 cells were co-transfected with Ce3-FRB-CEP170C, and a yellow fluorescent protein (Neon)-tagged FKBP (Neon-FKBP) (Fig. 1B). Before rapamycin treatment, Ce3-FRB-CEP170C localized at centrosomes, whereas Neon-FKBP remained in the cytosol (Fig. 1B). The addition of rapamycin triggered the accumulation of cytosolic Neon-FKBP at centrosomes (Fig. 1B,C; Movie EV1) with a half-time of $17.54 \pm 3.30$ s (Fig. 1D). We also attempted to recruit POIs to centrosomes using one of the rapamycin orthogonal CID systems that utilizes the plant hormone gibberellin to trigger dimerization between GAIs (N-terminal fragment of gibberellin insensitive) and mGID1 (codon-optimized gibberellin insensitivity DWARF1) (Lin et al, 2013; Miyamoto et al, 2012). Addition of the gibberellin analog GA3-AM resulted in the rapid accumulation of Neon-mGID1 at centrosomes where GAIs-CFP-CEP170C was localized (Appendix Fig. S4; Movie EV2; $T_{1/2} = 43.20 \pm 12.58$ s). These results confirmed that the CID system enables the rapid recruitment of POIs to centrosomes in living cells within seconds.

We previously engineered the deglutamylase CCP5 by tagging FKBP with the CCP5 catalytic domain (CCP5CD). Accumulation of the resulting enzyme fusion (CCP5CD-Neon-FKBP) to ciliary axonemes spatiotemporally depleted axonemal glutamylation (Hong et al, 2018). Leveraging this approach, we found that rapamycin treatment resulted in the recruitment of CCP5CD-Neon-FKBP to Ce3-FRB-CEP170-positive centrosomes (Fig. 1E). This accumulation significantly reduced centrosomal glutamylation by ~50% within 30 min of CID induction (Fig. 1E,F). Recruitment of an inactive enzyme, CCP5CDDM-Neon-FKBP (Hong et al, 2018), to centrosomes did not perturb centrosomal glutamylation,

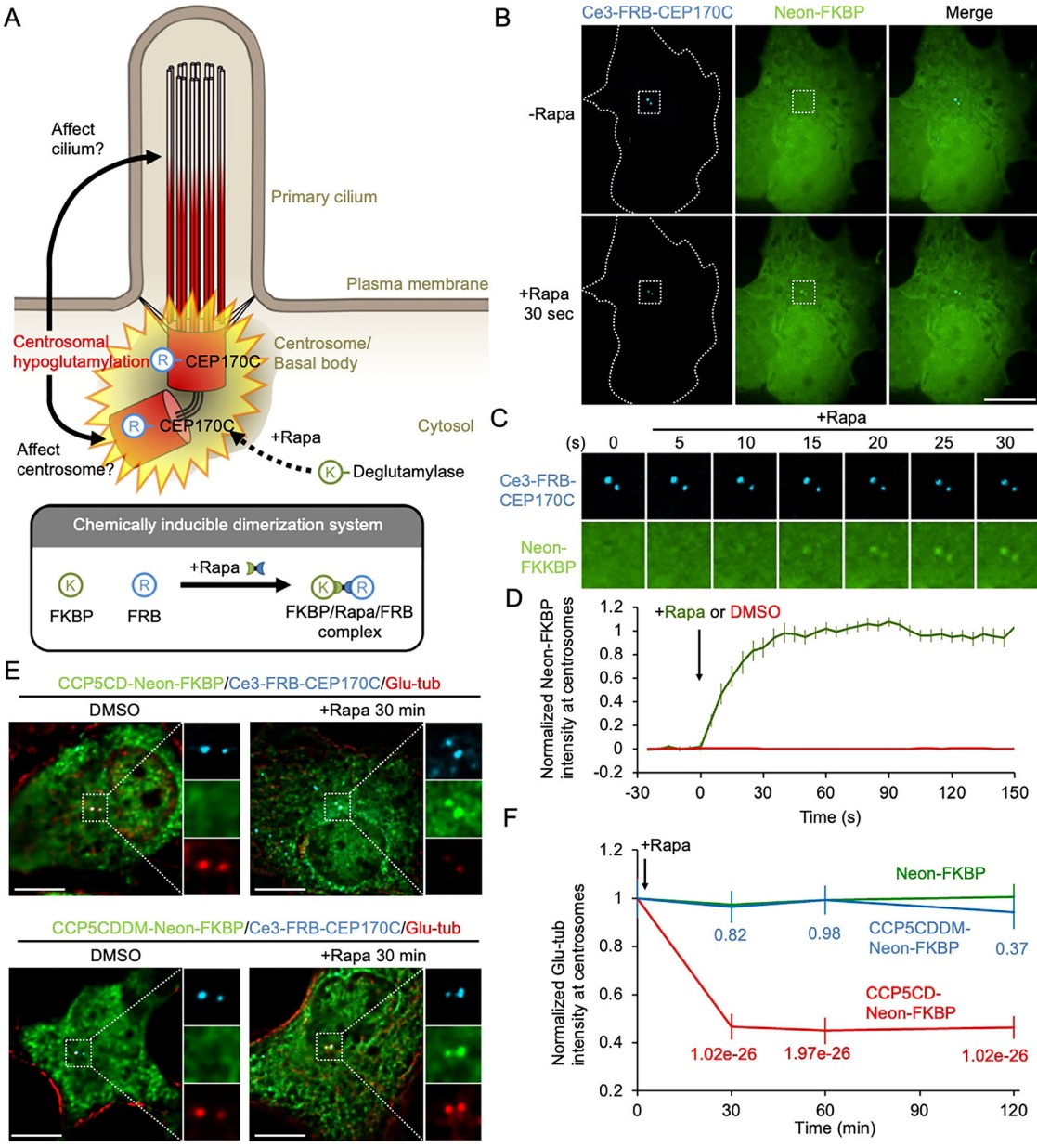

**Figure 1. Rapid recruitment of a deglutamylase to centrosomes for reducing centrosomal glutamylation.**

(A) Schematic diagram of our newly developed approach, which recruits an engineered deglutamylase from the cytosol to centrosomes via a chemically inducible dimerization system. The cytosolic FKBP-tagged deglutamylase is recruited to the FRB-tagged centrosomal targeting protein (CEP170C) upon rapamycin treatment (Rapa). The study evaluates the physiological roles of centrosomal glutamylation in ciliogenesis, as well as the structural integrity and functions of centrosomes. (B) Rapid recruitment of FKBP-tagged proteins from the cytosol to centrosomes. COS7 cells co-transfected with Cerulean3 (Ce3)-FRB-CEP170C and Neon-FKBP were treated with rapamycin (Rapa; 100 nM) for 30 s. Dashed lines indicate the cell boundary. Scale bar, 10 μm. (C) Video frames of the centrosome region highlighted in the dashed squares in (B). (D) Normalized fluorescence intensity of Neon-FKBP at centrosomes upon treatment with DMSO control (0.1%; red) or rapamycin (Rapa, 100 nM; green). Data represent the mean ± SEM. $n = 21$ and 24 cells in the DMSO and rapamycin groups, respectively, from four independent experiments. The arrow indicates the time of treatment. (E) COS7 cells were co-transfected with Ce3-FRB-CEP170C and either CCP5CD-Neon-FKBP or the enzymatically inactive mutant, CCP5CDDM-Neon-FKBP. After 24 h of transfection, cells were treated with 0.1% DMSO or rapamycin (Rapa; 100 nM) for 30 min, followed by immunostaining with GT335 antibody (red). The right panels show magnified images of the areas outlined by the dashed squares. Scale bar, 10 μm. (F) Normalized density of glutamylated tubulin at centrosomes after rapamycin-induced translocation of Neon-FKBP (green), CCP5CD-Neon-FKBP (red) or CCP5CDDM-Neon-FKBP (blue) for the indicated times. Data represent the mean ± SEM. ($n = 811$, 725, and 674 cells for the Neon-FKBP, CCP5CD-Neon-FKBP, and CCP5CDDM-Neon-FKBP groups, respectively, from three to five independent experiments). The arrow indicates the time of rapamycin treatment. Student's $t$ tests were performed, and the corresponding $P$ values are indicated. Source data are available online for this figure.

indicating this inducible hypoglutamylation at centrosomes depended on the enzyme activity of CCP5CD (Fig. 1E,F). No difference in cytosolic glutamylation level was observed, confirming that the reduction in glutamylation was specific to centrosomes (Fig. EV1A). Furthermore, this centrosomal hypoglutamylation occurred uniformly at both mother and daughter centrioles, as no significant difference in tubulin glutamylation levels was detected between the two centrioles (Fig. EV1B). Due to the irreversibility of CID (Lin et al, 2013; Voß et al, 2015; Liu et al, 2022; Chen et al, 2024), the effect of centrosomal hypoglutamylation was sustained for up to 24 h (Fig. EV1C). Additionally, this strategy was successfully applied to NIH3T3 fibroblast and U2OS epithelial cells (Fig. EV1D), confirming its broad applicability. Consistent with previous studies (Wang et al, 2023; Hong et al, 2018), CCP5CD preferentially reduced GT335-labeled glutamylated tubulin while having minimal impact on long-chain polyglutamated tubulin detected by PolyE antibody (Appendix Fig. S5). In summary, our results demonstrated that the CID-mediated recruitment of CCP5CD to CEP170C-labeled centrosomes enables rapid, specific, and effective centrosomal glutamylation reduction.

## Centrosomal hypoglutamylation has minimal impact on the centrosomal structure

Glutamylation is predominantly enriched in stable subpopulations of microtubule polymers (Janke and Magiera, 2020), suggesting that glutamylation may contribute to microtubule stability. To address this, the morphology of different centrosomal subcompartments, including centrioles, pericentriolar matrix, distal appendages, and subdistal appendages, was monitored before and after centrosomal glutamylation reduction. In both the CCP5CD and control groups, the morphology of centrioles and pericentriolar matrix remained unchanged (Fig. 2A,B,F,G). Moreover, distal appendages (labeled by CEP83 and CEP164) and subdistal appendages (labeled by ODF2) retained normal morphology after 30 min of centrosomal glutamylation reduction (Fig. 2C–E,H–J). 3D-SIM was utilized to examine and quantify centriolar acetylation intensity as well as the length of acetylated tubulin-labeled centrioles in a longitudinal orientation before and after glutamylation reduction. The results confirmed that hypoglutamylation does not significantly alter centriole acetylation or length (Fig. 2K–M). To induce long-term centrosomal hypoglutamylation, CCP5CD was targeted to centrosomes by tagging it with CEP170C. Expression of CCP5CD-mCh-CEP170C did not affect the density of Centrin2, CEP164, or Pericentrin when compared to control conditions (Appendix Fig. S6A–F). Collectively, these results indicate that both acute and long-term centrosomal hypoglutamylation have minimal impact on centrosome structure.

## Centrosomal glutamylation promotes microtubule nucleation

Besides structure, we next comprehensively evaluated the effects of hypoglutamylation on centrosomal functions, particularly its role in regulating microtubule nucleation (Doxsey et al, 2005). To assess this, we performed a microtubule regrowth assay comparing normal and hypoglutamylated centrosomes. Pre-existing microtubules were first disassembled to soluble tubulin monomers by treating cells with nocodazole at 4 °C. Nascent microtubules then

regrew from centrosomes once the cells were returned to physiological conditions in the absence of nocodazole (O'Rourke et al, 2014). In CCP5CD-transfected cells, microtubule regrowth from rapamycin-induced hypoglutamylated centrosomes was significantly reduced compared to centrosomes treated with vehicle (dimethylsulfoxide, DMSO), whereas microtubule regrowth did not differ significantly in the likewise-treated CCP5CDDM group (Fig. 3A,B).

To assess the real-time impact of centrosomal hypoglutamylation on microtubule growth, we tagged EB1 (End Binding Protein 1) with YFP. EB1 localizes to the plus ends of elongating microtubules in living cells (Salaycik et al, 2005). The number of EB1-YFP comets emanating from centrosomes was monitored and quantified via live-cell imaging (Fig. 3C,D). CCP5CD-mediated hypoglutamylation at centrosomes significantly reduced the number of centrosome-derived EB1-YFP comets compared to Neon-FKBP and CCP5CDDM-Neon-FKBP-transfected cells, indicating that acute centrosomal glutamylation reduction immediately impairs microtubule growth (Fig. 3C,D). Furthermore, long-term centrosomal hypoglutamylation also led to a decrease in the number of EB1-tracks originating from centrosomes (Appendix Fig. S6G,H). These results collectively demonstrated that glutamylation at centrosomes positively regulates microtubule nucleation.

## Centrosomal glutamylation physically recruits CEP192/NEDD1/γ-tubulin complexes for microtubule nucleation via electrostatic forces

Pericentrin is a known scaffold protein that organizes pericentriolar architecture and facilitates microtubule nucleation (Delaval and Doxsey, 2010). The minimal impact of centrosomal hypoglutamylation on pericentrin suggests that glutamylation regulates microtubule nucleation through pericentrin-independent mechanisms (Fig. 2B,G; Appendix Fig. S6C,F). γ-tubulin is a key component of the γ-tubulin ring complex that acts as the major machinery that nucleates microtubule in cells (Sulimenko et al, 2017). Moreover, CEP192 and NEDD1 (neural precursor cell–expressed developmentally downregulated protein 1) anchor γ-tubulin and the ring complex to centrosomes (Haren et al, 2006; Lüders et al, 2006; Joukov et al, 2014). To unravel the molecular mechanisms of how centrosomal glutamylation regulates microtubule nucleation, we attempted to measure the density of γ-tubulin, CEP192, and NEDD1 at centrosomes upon the CCP5CD-mediated hypoglutamylation. Immunostaining revealed that centrosomal hypoglutamylation diminished the recruitment of γ-tubulin, CEP192, and NEDD1 to centrosomes, an effect not observed in the Neon-FKBP alone and CCP5CDDM-Neon-FKBP groups (Fig. 4A–F). These results demonstrated that glutamylation recruits microtubule nucleation factors such as CEP192, NEDD1, and γ-tubulin to centrosomes.

Since CEP192 is known to physically recruit the NEDD1/γ-tubulin complex (Gomez-Ferreria et al, 2012a, 2012b), we next assessed whether hypoglutamylation-induced γ-tubulin loss and defective microtubule nucleation depend on CEP192. To address this, exogenous CEP192 was constrained at centrosomes by tagging it with CEP170C, allowing us to determine whether the resulting chimeric protein could rescue the defects of γ-tubulin-mediated microtubule nucleation under centrosomal hypoglutamylation. The CEP192-Ce3-FRB-CEP170C construct constrained CEP192 at centrosomes and recruited CCP5CD-Neon-FKBP to centrosomes

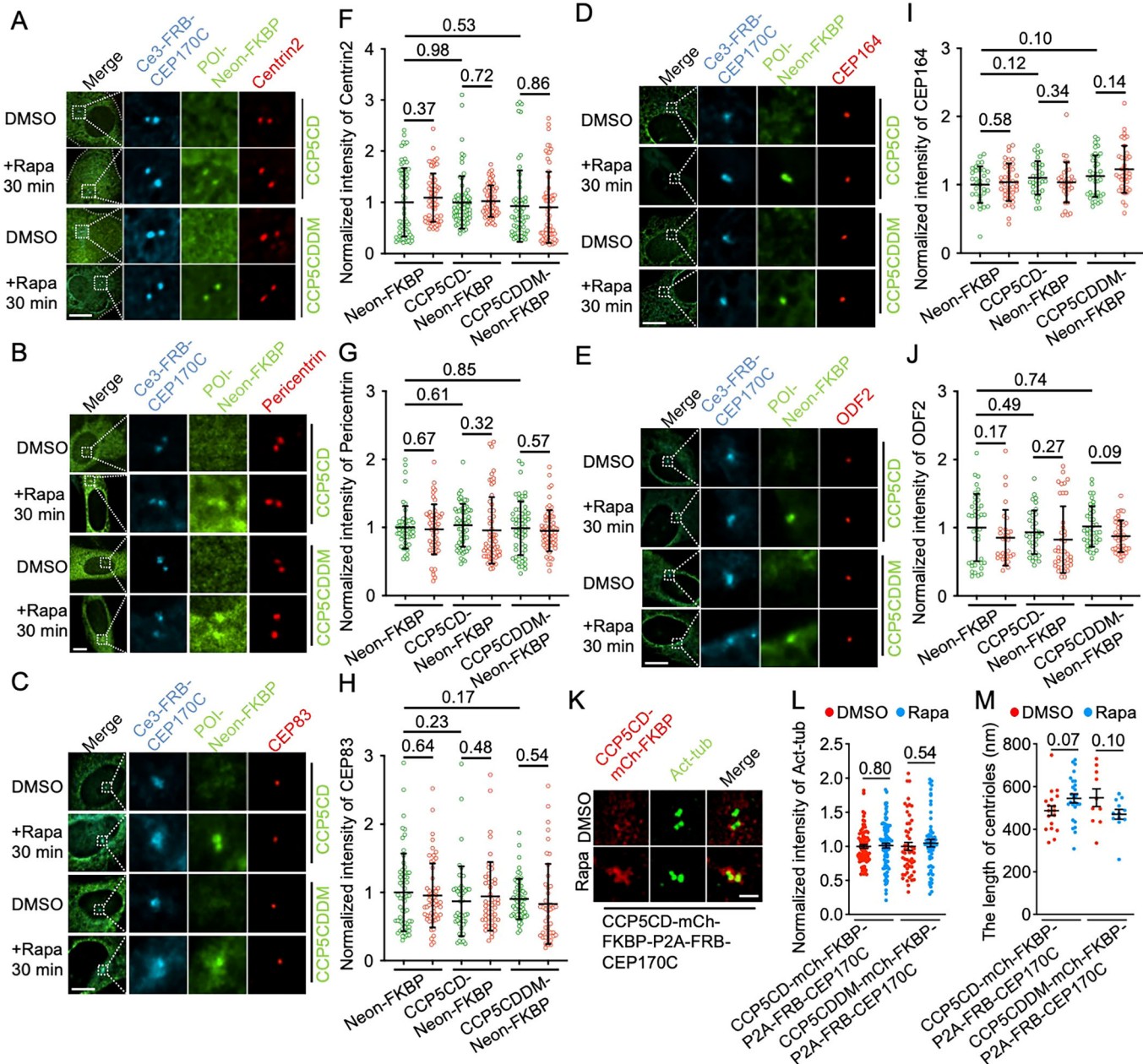

**Figure 2. Centrosomal hypoglutamylation has minimal impact on centrosome structure.**

(A–E) NIH3T3 cells co-transfected with Ce3-FRB-CEP170C and the indicated constructs were treated with either 0.1% DMSO or rapamycin (Rapa, 100 nM) for 30 min and then immunostained with antibodies against Centrin2 (A), Pericentrin (B), CEP83 (C), CEP164 (D), and ODF2 (E). The right panels show magnified images of the areas outlined by the dashed squares. Scale bar, 10 μm. (F–J) Normalized density of the indicated centrosome proteins after 0.1% DMSO (green) or rapamycin (100 nM; red) treatment in cells transfected with the indicated constructs (A–E). Data represent the mean ± SD. (n = 388 (F), 343 (G), 311 (H), 210 (I), and 224 cells (J) from three to five independent experiments). (K). 3D-SIM images of centrosomes in COS7 cells transfected with CCP5CD-mCh-FKBP-P2A-FRB-CEP170C upon 0.1% DMSO or rapamycin treatment. Cells were immunostained with antibodies against mCherry (mCh; red) and acetylated tubulin (act-tub; a marker of centrioles; green). Scale bar, 1 μm. (L, M) Normalized intensity of acetylated tubulin (L) and centriole length (M) in cells from (K) after 30 min of 0.1% DMSO (red) or 100 nM rapamycin (blue) treatment. Data represent as the mean ± SEM. n (from left to right) = 80, 93, 53, and 60 cells in (L); 18, 26, 10, and 12 cells in (M); from three independent experiments. Student's t tests were performed, and P values are indicated. Source data are available online for this figure.

for inducible glutamylation reduction. Compared with cells expressing Ce3-FRB-CEP170C, those transfected with CEP192-Ce3-FRB-CEP170C exhibited restored centrosomal NEDD1 and γ-tubulin density (Fig. EV2A–D), suggesting that the defect in γ-tubulin-mediated microtubule nucleation at hypoglutamylated

centrosomes is caused by impaired centrosomal anchoring of CEP192. In addition to CEP192, constraining NEDD1 to hypoglutamylated centrosomes also rescued defects in γ-tubulin-mediated microtubule nucleation (Fig. 4G–J), indicating that NEDD1 is also critical for this process.

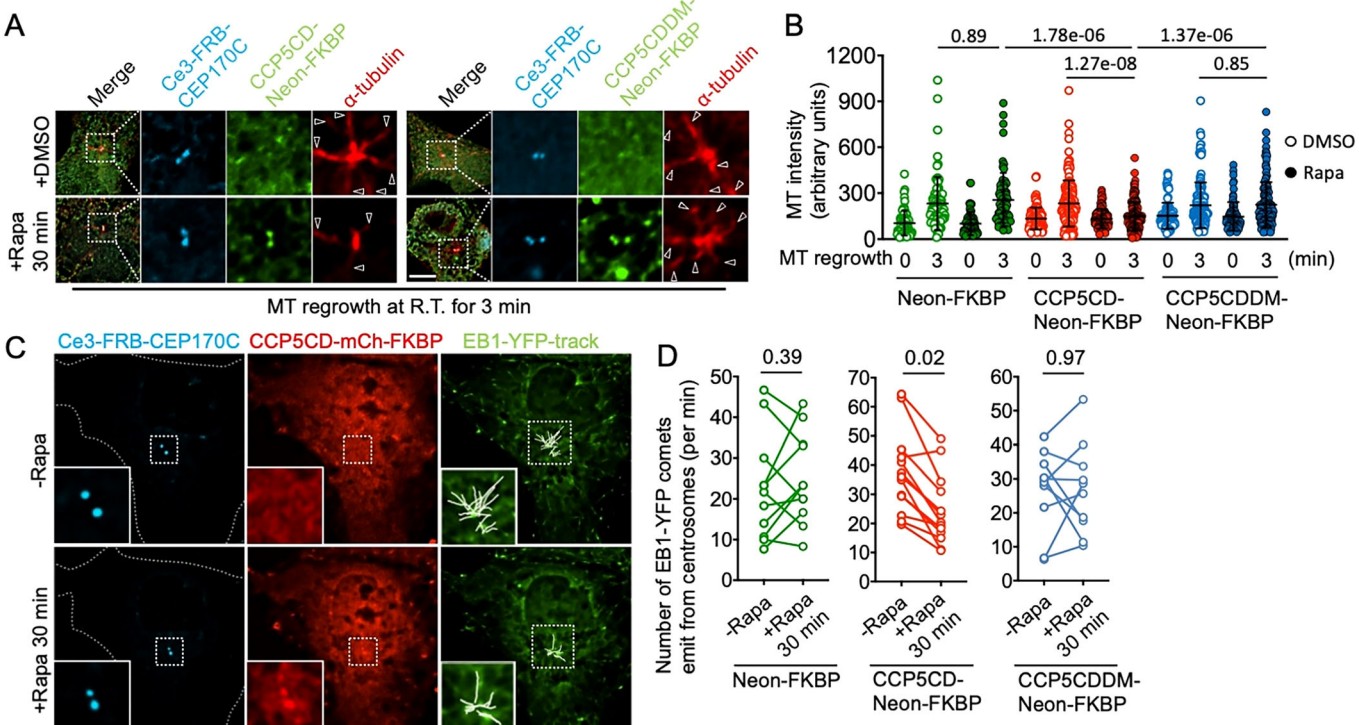

**Figure 3. Centrosomal glutamylation is essential for microtubule nucleation.**

(A–D) COS7 cells were co-transfected with Ce3-FRB-CEP170C and either CCP5CD-Neon-FKBP or CCP5CDDM-Neon-FKBP. At 80–90% confluency, transfected cells were treated with 100 nM rapamycin (Rapa) or 0.1% DMSO for 30 min and then placed on ice for 40 min to depolymerize microtubules. Cells were subsequently washed with DMEM for 1 min, allowed to recover for 3 min at room temperature (RT), and then fixed for immunostaining with anti-α-tubulin antibody. The right panels show magnified images of the areas outlined by the dashed squares. Hollow arrowheads indicate regrown microtubules, as shown by α-tubulin-labeled filaments extending from the centrosomal region. Scale bar, 10 μm. (B) Quantification of centrosome-derived microtubule intensity with or without 3 min recovery in cells transfected with the indicated constructs. Data represent the mean ± SEM (*n* = 270, 499, and 439 cells for the Neon-FKBP (green), CCP5CD-Neon-FKBP (red), and CCP5CDDM-Neon-FKBP groups (blue), respectively, from four to five independent experiments). (C) COS7 cells were co-transfected with Ce3-FRB-CEP170C, CCP5CD-mCh-FKBP, and EB1-YFP. One day post-transfection, EB1-YFP-positive comets were monitored by live-cell imaging in the same cell before and after rapamycin-induced centrosomal hypoglutamylation. Centrosomal microtubule tracks were drawn based on EB1-YFP time-lapse imaging with 2 min duration. Insets show higher-magnification images of the centrosomal regions. Scale bar, 10 μm. (D) Quantification of EB1-YFP comet frequency emitted from centrosomes in cells co-transfected with Ce3-FRB-CEP170C and the indicated constructs before and after treatment with rapamycin (100 nM) for 30 min. *n* = 11, 13 and 10 cells in the Neon-FKBP, CCP5CD-Neon-FKBP and CCP5CDDM-Neon-FKBP groups, respectively, from four to five independent experiments. Student's *t* tests were performed, and *P* values are indicated. Source data are available online for this figure.

Since the mechanism of NEDD1 binding to centrosomes has been well-studied (Haren et al, 2006; Manning and Kumar, 2007), we next aimed to explore how glutamylation recruits NEDE1 to centrosomes in greater detail. We used a co-sedimentation assay to test whether glutamylation promotes the physical interaction between microtubules and NEDD1. To general and purify glutamylated microtubules, we expressed TTLL4C1-myc, a truncated form of TTLL4 that induces microtubule hyperglutamylation (van Dijk et al, 2007), in human embryonic kidney (HEK293T) cells (Fig. 4K). Overexpression of TTLL4C1 increased microtubule glutamylation by 4.51-fold (Fig. 4L). Immunoblotting demonstrated that glutamylated microtubules co-sedimented with NEDD1 to a greater extent than with the unmodified microtubules (2.65 ± 0.46-fold difference) (Fig. 4L,M). Moreover, 3D-SIM images showed that NEDD1 partially overlapped with glutamylated tubulin on the surface of centrioles (Fig. 4N). In summary, these results indicate that glutamylation contributes to the physical interaction between centrioles and NEDD1.

NEDD1 anchors to centrioles via its N-terminal region (Haren et al, 2006), which is highly conserved among different species and has a positively charged surface (charge: +9.56; Fig. 4O; Appendix Fig. S7A). Because glutamylated side chains are negatively charged, we hypothesized that the electrostatic forces may drive the interaction between NEDD1 and glutamylated centrioles (Guichard et al, 2023). To test this hypothesis, the conserved, positively charged amino acids residue on the surface of the NEDD1 N-terminal centrosome-binding domain were replaced with negatively charged residues (Appendix Fig. S7A). The resulting "negative NEDD1" had a net charge of −11.44, with minimal alterations in its overall structure (0.013 Å; Fig. 4O; Appendix Fig. S7B). As expected, wild-type NEDD1 localized to centrosomes (Fig. 4P–R) (Haren et al, 2006). However, negative NEDD1 resided in the cytosol and failed to anchor to centrosomes (Fig. 4R), suggesting that NEDD1 physically interacts with glutamylated centrioles via electrostatic forces. Notably, constraining negative NEDD1 at the centrosomes through CEP170C tagging still allowed

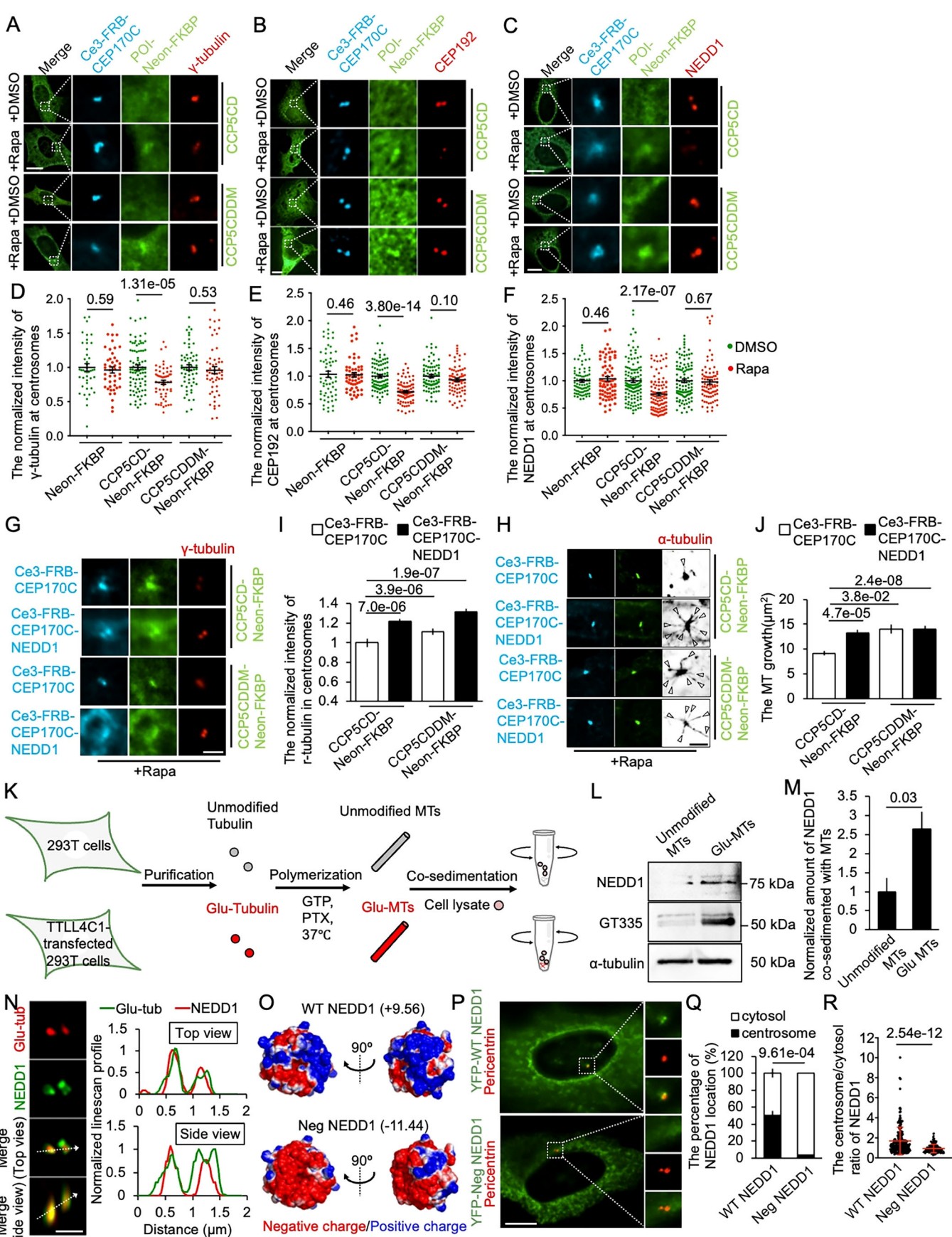

**Figure 4.  Glutamylation physically recruits NEDD1 and γ-tubulin via electrostatic forces for microtubule nucleation.**

(A–C) NIH3T3 cells co-transfected with Ce3-FRB-CEP170C and either CCP5CD-Neon-FKBP or CCP5CDDM-Neon-FKBP were treated with DMSO (0.1%) or rapamycin (Rapa, 100 nM) for 1 h. Following treatment, cells were fixed and immunostained for γ-tubulin (A), CEP192 (B), and NEDD1 (C), respectively. The right panels show magnified images of the areas outlined by dashed squares. Scale bar, 10 µm. (D–F) Quantification of the normalized fluorescence intensity of γ-tubulin (D), CEP192 (E), and NEDD1 (F) in cells from (A), (B), and (C), respectively. Data represent the mean ± SEM. n (Neon-FKBP, CCP5CD-Neon-FKBP, and CCP5CDDM-Neon-FKBP) = 87, 142, and 113 cells in (D); 117, 174, and 159 cells in (E); 153, 229, and 192 cells in (F), respectively, from three to four independent experiments. (G, H) NIH3T3 cells were co-transfected with Ce3-FRB-CEP170C and CCP5CD-Neon-FKBP or with Ce3-FRB-CEP170C-NEDD1 and CCP5CDDM-Neon-FKBP. In (G), transfected cells were treated with 100 nM rapamycin (Rapa) for 1 h and then immunostained for γ-tubulin. In (H), transfected cells were incubated on ice with nocodazole (3.3 µM) for 40 min to depolymerize microtubules. Following depolymerization, cells were washed with DMEM for 1 min, allowed to recover for 3 min at RT, and then immunostained with anti-α-tubulin antibody. Hollow arrowheads indicate regrown microtubules, as shown by α-tubulin-labeled filaments extending from the centrosomal region in (H). Scale bar, 2 µm. (I, J) Quantification of the normalized intensity of γ-tubulin in (I) and area of centrosome-derived microtubules in (J) are shown. Data represent the mean ± SEM. n = 212 and 222 cells in CCP5CD and CCP5CDDM groups, respectively, for (I); n = 158 and 149 cells in CCP5CD and CCP5CDDM groups, respectively, for (J); three independent experiments. (K) Schematic of the co-sedimentation experiment protocol. Purified glutamylated or unmodified microtubules were generated and incubated with NIH3T3 cell lysates. The lysate proteins co-sedimented with microtubules (MTs) were collected and subjected to immunoblotting. (L) Immunoblot analysis of cell lysates collected using the protocol in (K), probing for NEDD1, glutamylated tubulin, and α-tubulin. (M) Quantification of NEDD1 levels co-sedimented with unmodified or glutamylated microtubules. Data represent the mean ± SEM from three independent experiments. (N) 3D-SIM images of tubulin glutamylation (Glu-tub) and NEDD1 in COS7 cells. The distribution profiles of tubulin glutamylation and NEDD1 are shown from both top view and side view of centrosomes. Scale bar, 1 µm. (O) Charge potential analysis of the centrosome-binding domain of wild-type (WT) NEDD1 and the negatively charged NEDD1 mutant (Neg). (P) COS7 cells transfected with WT NEDD1 or Neg NEDD1 were immunostained for pericentrin (red). The right panels show magnified images of the areas outlined by dashed squares. Scale bar, 10 µm. (Q, R) Quantification of the percentage of cells with centrosomal or cytosolic localization of NEDD1 (Q) and the centrosome-to-cytosol intensity ratio of NEDD1 (R). Data represent the mean ± SEM in (Q) and mean ± SD (red) in (R). n = 200 and 204 cells in the WT and Neg NEDD1 groups, respectively; Three independent experiments. Student's t tests were performed, and P values are indicated. Noted: Single focal plane of Ce3-FRB-CEP170C images may not always display a two-centriole pattern. Source data are available online for this figure.

γ-tubulin recruitment (Appendix Fig. S8). Haren et al, demonstrated that the N-terminus and C-terminus of NEDD1 protein are responsible for distinct functions: centrosome binding and γ-tubulin interaction, respectively (Haren et al, 2006). In line with this, our findings suggest that introducing negatively charged mutations into the N-terminus of NEDD1 affects its centrosome binding but does not impact its interaction with γ-tubulin. Taken together, our results demonstrated that centrosomal glutamylation physically anchors CEP192 and NEDD1 protein complex via electrostatic forces, which in turn recruits γ-tubulin to facilitate microtubule nucleation.

## Centrosomal hypoglutamylation perturbs the distribution of centriolar satellites

Centriolar satellites are cytosolic granules that are scattered around centrosomes and continuously shuttle along microtubules to deliver cargoes to centrosomes, thereby ensuring proper centriole signaling and ciliogenesis (Aydin et al, 2020; Hori et al, 2016; Odabasi et al, 2019; Wang et al, 2016). Microtubule depolymerization perturbs the distribution of centriolar satellites (Dammermann et al, 2004; Dammermann and Merdes, 2002). We next attempted to evaluate the effects of centrosomal hypoglutamylation on the distribution and trafficking of the centriolar satellites. One of the major components of these satellites, PCM1 (pericentriolar material 1), was used to mark centriolar satellites in NIH3T3 cells. Consistent with the previous observation, PCM1-positive centriolar satellites were abundant in the vicinity of centrosomes (Fig. 5A) (Hori et al, 2016). However, acute centrosome glutamylation reduction significantly altered the distribution of the satellites, as evidenced by the condensed and dispersed patterns of PCM1-positive granules (Fig. 5A,B). This aberrant distribution did not affect the PCM1 level (Fig. 5C). We also quantified the condensed centriolar satellites by measuring the area of PCM1-positive granules around centrosomes, which was reduced upon acute centrosome glutamylation reduction. This indicated that centriolar satellites were sequestered

at centrosomes after glutamylation reduction (Fig. 5D). Recruitment of Neon-FKBP alone or CCP5CDDM-Neon-FKBP to centrosomes had no effect on PCM1 distribution, indicating that the defect in PCM1 distribution depended on the enzyme activity of deglutamylases (Fig. 5A–D). We further monitored the real-time dynamics of centriolar satellites upon centrosome hypoglutamylation. A F2 fragment of PCM1 (residues 590–1460; hereafter PCM1F2) was tagged with mCherry to visualize centriolar satellites in living cells (Wang et al, 2013; Cheng et al, 2022). The PCM1F2-mCh-positive centriolar satellites gradually accumulated at hypoglutamylated centrosomes (Fig. 5E; Movie EV3). These results confirmed that centrosomal glutamylation is continuously required for proper centriolar satellite distribution. We next evaluated whether NEDD1 contributes to centriolar satellite defects caused by centrosomal hypoglutamylation. Rescue experiments showed that the centrosomal recruitment of exogenous Ce3-FRB-CEP170C-NEDD1 increased the percentage of cells exhibiting normal centriolar satellite distribution and restored the centriolar satellites area to the baseline levels—even after centrosomal glutamylation reduction (Fig. 5F–H). Taken together, these results confirmed that centrosomal glutamylation regulates the distribution of centriolar satellites via a NEDD1-dependent pathway.

## Centrosomal glutamylation ensures ciliogenesis and cilia maintenance

The centrosome transforms to a basal body during the G0 cell-cycle phase, serving as a platform to extend the primary cilium (Bernabé-Rubio and Alonso, 2017). Centriolar satellites transport ciliogenesis factors between centrioles/basal bodies and the cytosol (Odabasi et al, 2020, 2019). Therefore, we next investigated whether glutamylation at basal bodies contributes to ciliogenesis or cilia maintenance. After acute centrosomal glutamylation reduction, cells were subsequently serum-starved for 4 h to induce ciliogenesis (Fig. 6A). After rapamycin treatment, the nascent primary cilia, labeled with anti-Arl13b, were significantly shorter in the CCP5CD

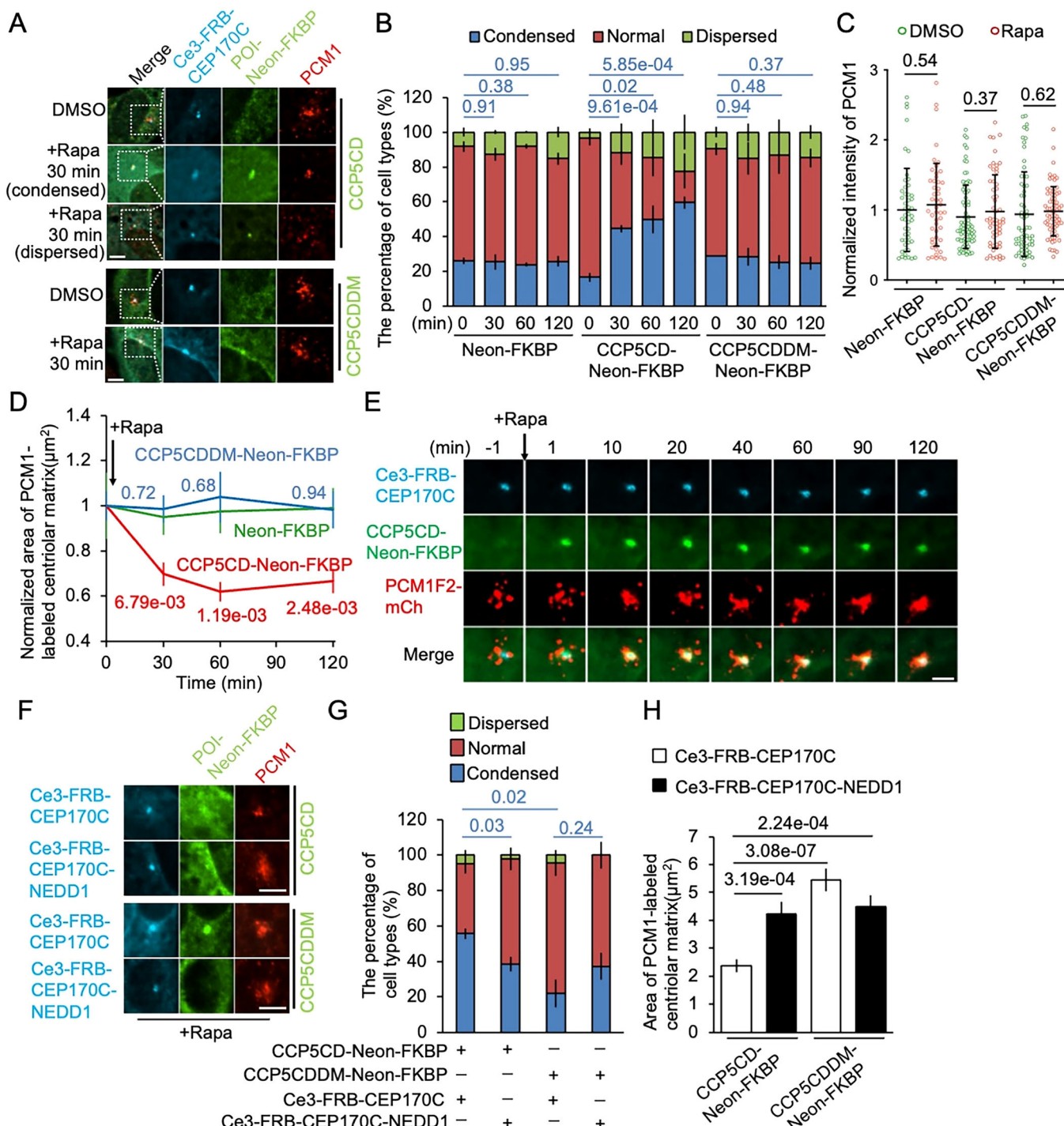

group compared to the Neon-FKBP alone and CCP5CDDM-Neon-FKBP groups (Fig. 6B,C). Moreover, serum starvation-induced ciliation was inhibited in hypoglutamylated basal bodies but remained unaffected in control groups (Fig. 6D). These defects in ciliogenesis persisted for up to 24 h (Fig. 6B–D). The results indicated that hypoglutamylation at centrosomes/basal bodies attenuates ciliogenesis (Fig. 6B–D). Next, we evaluated the impact of basal-body hypoglutamylation on the maintenance of mature cilia. To address this, we first induced cilia formation by 24 h of serum starvation, followed by acute basal-body hypoglutamylation (Fig. 6E). Although cilia length remained unchanged within the first 30 min of basal-body hypoglutamylation, it began to decrease after 1 h of glutamylation reduction in the CCP5CD group, but not in the Neon-FKBP alone and CCP5CDDM-Neon-FKBP groups (Fig. 6F,G). While hypoglutamylation led to a reduction in cilia length, it did not affect overall population of ciliated cells (Fig. 6H). Furthermore, we demonstrated that long-term basal-body hypoglutamylation attenuated cilia formation (Appendix Fig. S6I–K).

**Figure 5. Centrosomal deglutamylation perturbs the dynamics and distribution of centriolar satellites via NEDD1.**

(A) NIH3T3 cells co-transfected with the indicated constructs were treated with DMSO (0.1%) or rapamycin (100 nM) for 30 min. Following treatment, cells were fixed and immunostained for PCM1. Representative images show condensed and dispersed PCM1 patterns in CCP5CD-Neon-FKBP-transfected cells. The right panels show magnified images of the areas outlined by dashed squares. Scale bar, 5 μm. (B) Percentage of cells exhibiting the indicated PCM1 pattern from (A). Data represent the mean ± SEM. $n = 1035$, 385, and 260 cells in Neon-FKBP, CCP5CD-Neon-FKBP, and CCP5CDDM-Neon-FKBP groups, respectively, from three independent experiments. (C) Normalized intensity of PCM1 in (A). Data represent the mean ± SD. $n = 99$, 139, 137 cells in Neon-FKBP, CCP5CD-Neon-FKBP, and CCP5CDDM-Neon-FKBP groups, respectively, from three independent experiments. (D) Quantification of the normalized area of PCM1-positive centriolar satellites around centrosomes in (A). Data represent the mean ± SEM. $n = 216$, 391, and 310 cells in the Neon-FKBP, CCP5CD-Neon-FKBP and CCP5CDDM-Neon-FKBP groups, respectively, from three to five independent experiments. (E) Video frames of NIH3T3 cells transfected with the indicated constructs upon rapamycin treatment (100 nM). Scale bar, 5 μm. (F) NIH3T3 cells co-transfected with the indicated constructs were treated with rapamycin (100 nM) for 1 h and immunostained for PCM1. Scale bar, 5 μm. (G) Percentage of cells exhibiting the indicated PCM1 pattern in (F). Data represent the mean ± SEM. $n = 45$, 56, 39, and 52 cells from left to right. Three independent experiments. (H) Quantification of the PCM1-positive centriolar satellite area around centrosomes in (E). Data represent the mean ± SEM. $n = 42$, 53, 36, and 50 cells from left to right. Three independent experiments. Student's $t$ tests were performed, and $P$ values are indicated. Source data are available online for this figure.

Hypoglutamylation at basal bodies also reduced axonemal glutamylation (Appendix Fig. S9). However, since relative axoneme glutamylation level (axonemal glutamylation/cilia length) remained comparable between glutamylated and hypoglutamylated basal bodies, we inferred that the lower axonemal glutamylation level resulted from the shorter cilia growing from hypoglutamylated basal bodies. Taken together, these results confirm that glutamylation at basal bodies is important for both ciliogenesis and cilia maintenance.

## Glutamylation at basal bodies is dispensable for the docking and entry of the intraflagellar machinery into cilia

Intraflagellar transport (IFT) is a motor protein-dependent transport mechanism that delivers structural and signaling cargoes to ensure proper cilia architecture and function (Hao and Scholey, 2009). The IFT machinery assembles at basal bodies and then transports cargo along ciliary axonemes (Yang et al, 2013). A previous in vitro study found that chemically conjugated glutamate chains on purified microtubules increases the progressivity and mobility of one IFT motor, kinesin-2 (Sirajuddin et al, 2014). We therefore hypothesized that glutamylation at basal bodies may facilitate the anchoring of the IFT machinery at basal bodies as well as the IFT pathway to support cilia structure. The IFT machinery was labeled with one of the IFT components, IFT88. Consistent with the results of previous studies (Yang et al, 2019; Hong et al, 2018), IFT88 was concentrated at the ciliary base (Appendix Fig. S10A, open arrowheads) and exhibited several puncta in cilia (Appendix Fig. S10A, filled arrowheads), confirming that the IFT machinery was anchored at the ciliary base and transported within cilia (Yang et al, 2019; Hong et al, 2018). However, although hypoglutamylation at basal bodies caused cilia to shorten (Fig. 6F; Appendix Fig. S6I–K), there was no significant difference in IFT88 density at the ciliary base or in cilia regardless of local deglutamylation at basal bodies (Appendix Fig. S10A,B). Fluorescence recovery after photobleaching (FRAP) was then utilized to monitor the impact of CCP5CD-mCh-CEP170C-mediated hypoglutamylation at basal bodies on the dynamics of Neon-IFT88 anchoring to basal bodies. After bleaching, IFT88 in both the CCP5CD and CCP5CDDM groups gradually reverted to basal bodies with akin kinetics (Appendix Fig. S10C,D). There was no significant difference in the half-time of recovery and the mobile fraction of IFT88 at basal bodies between cells transfected

with CCP5CD-mCh-CEP170C or CCP5CDDM-mCh-CEP170C (Appendix Fig. S10E). Consistent with a previous study (Van Den Hoek et al, 2022), we also noticed that the ciliary base, where the IFT88-positive machinery was anchored, did not colocalize with the glutamylated region of basal bodies (Appendix Fig. S11), implying that glutamylation does not directly contribute to IFT anchoring at the ciliary base. These results confirmed that glutamylation at basal bodies is dispensable for docking of the IFT88 machinery at basal bodies as well as its entry into cilia. It is unlikely that defects in ciliogenesis or cilia maintenance—as induced by basal-body hypoglutamylation—depend on the IFT pathway.

## Glutamylation regulates ciliogenesis by NEDD1 and Talpid3

Because centriolar satellites help regulate ciliogenesis, we hypothesized that centrosomal hypoglutamylation leads to defects in CEP192/NEDD1/γ-tubulin-mediated microtubule nucleation and centriolar satellite dynamics, which in turn perturbs ciliogenesis or cilia maintenance. To address this possibility, CEP170C-NEDD1 was expressed to rescue ciliogenesis in cells with hypoglutamylated centrosomes. Expression of CEP170C-NEDD1 triggered ciliogenesis even after glutamylation was reduced at basal bodies (Fig. 7A,B), indicating that NEDD1 is involved in glutamylation-mediated ciliogenesis.

Talpid3 is a key ciliogenesis factor whose expression is regulated by centriolar satellites (Wang et al, 2016). Thus, it was plausible that glutamylation could facilitate centriolar satellite trafficking to promote ciliogenesis via Talpid3. We found that acute reduction of glutamylation at basal bodies led to a decrease in Talpid3 level (Fig. 7C,D). Moreover, ectopic expression of CEP170C-Talpid3 rescued the ciliogenesis defects caused by basal-body hypoglutamylation (Fig. 7E,F). These results confirmed that glutamylation at basal bodies ensures ciliogenesis through NEDD1 and Talpid3.

## Centrosomal hypoglutamylation perturbs mitotic spindle formation and prolongs mitosis

Besides its role in the nucleation of interphase microtubules, NEDD1/r-tubulin signaling is also important for mitotic spindle assembly and proper cell division (Haren et al, 2006). We therefore assessed the effects of centrosomal hypoglutamylation on mitotic spindle formation. HeLa cells were synchronized in G2/M and

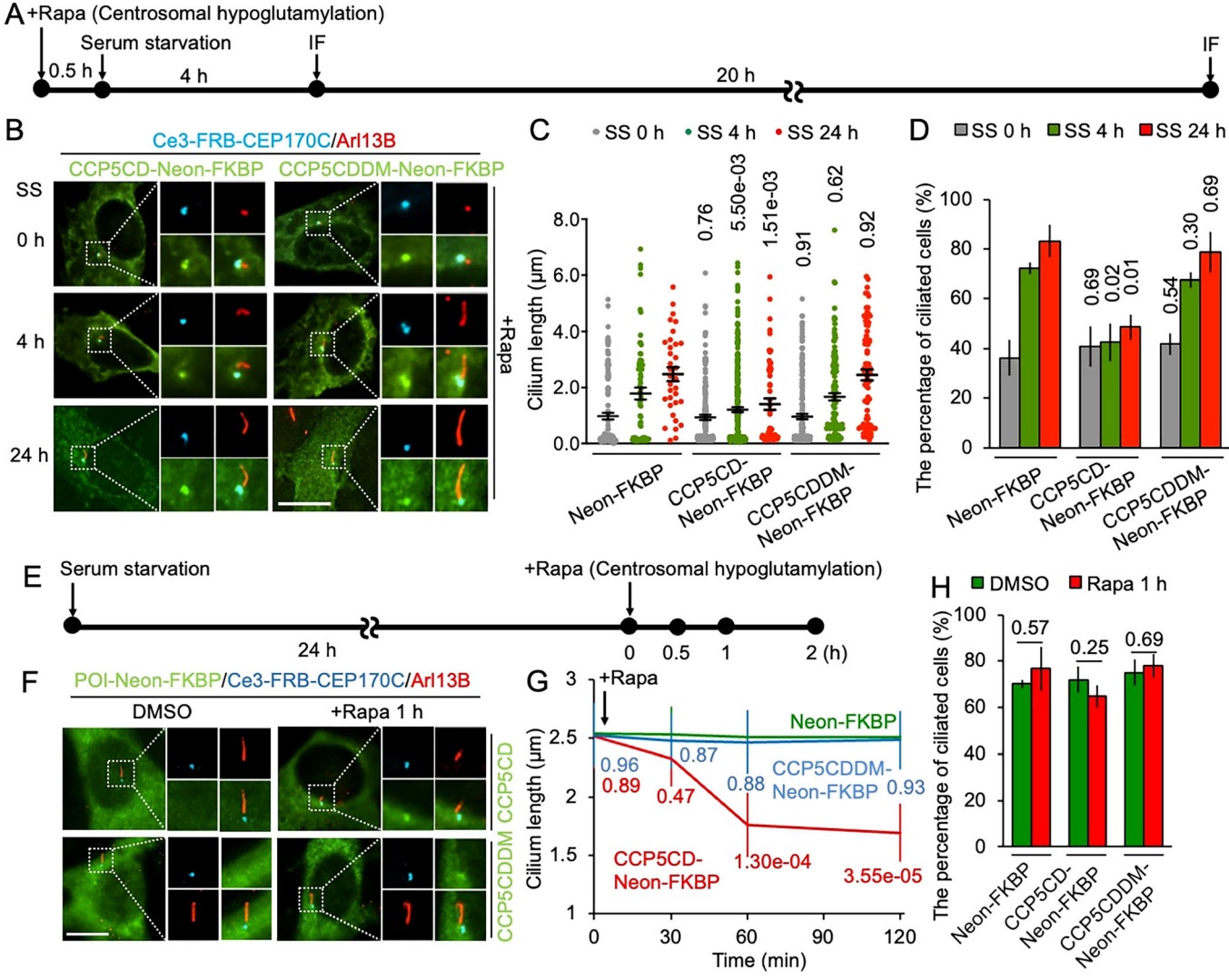

**Figure 6. Centrosomal glutamylation is required for ciliogenesis and cilia maintenance.**

(A) Experimental protocol to induce acute centrosomal hypoglutamylation followed by serum starvation–induced ciliogenesis. Cells were serum-starved for 4 h or 24 h, after which nascent and mature primary cilia were stained for immunofluorescence analysis (IF). (B) NIH3T3 cells co-transfected with Ce3-FRB-CEP170C and either CCP5CD-Neon-FKBP or CCP5CDDM-Neon-FKBP were serum-starved (SS) for the indicated times according to the protocol in (A). Cilia were labeled by anti-Arl13B antibody. The right panels show magnified images of the areas outlined by dashed squares. Scale bar, 10 μm. (C) Quantification of cilium length in cells from (B). Data (black) represent the mean ± SEM. n = 225, 474, and 392 cells in Neon-FKBP, CCP5CD-Neon-FKBP, and CCP5CDDM-Neon-FKBP groups, respectively, obtained from 8 to 10 independent experiments. (D) Percentage of ciliated cells in (B). Data represent the mean ± SEM. n = 534, 399, and 446 cells in Neon-FKBP, CCP5CD-Neon-FKBP, and CCP5CDDM-Neon-FKBP groups, respectively, obtained from three independent experiments. (E) Experimental protocol to induce the formation of mature cilia via 24 h of serum starvation, followed by rapamycin (Rapa)-induced basal-body hypoglutamylation for the indicated times. (F) NIH3T3 cells co-transfected with Ce3-FRB-CEP170C and either CCP5CD-Neon-FKBP or CCP5CDDM-Neon-FKBP were treated according to the protocol in (E). Cilia were labeled by anti-Arl13B antibody. The right panels show magnified images of the areas outlined by dashed squares. Scale bar, 10 μm. (G) Quantification of cilium length of cells from (F). Data represent the mean ± SEM. n = 520, 946 and 276 cells in Neon-FKBP, CCP5CD-Neon-FKBP, and CCP5CDDM-Neon-FKBP groups, respectively, obtained from six independent experiments. (H) Percentage of ciliated cells in (F). Data represent the mean ± SEM. n = 580, 1093, and 244 cells in Neon-FKBP, CCP5CD-Neon-FKBP, and CCP5CDDM-Neon-FKBP groups, respectively, obtained from three to seven independent experiments. Student's t tests were performed, and P values are indicated. Source data are available online for this figure.

released into mitosis immediately after centrosomal hypoglutamylation treatment (Fig. 8A). Most control cells exhibited bipolar mitotic spindles with properly aligned chromosomes at the metaphase plate. However, centrosomal hypoglutamylation resulted in aberrant mitotic spindles and altered chromosome distribution (Fig. 8B,C). Moreover, these defects in mitotic spindle formation and chromosome alignment significantly prolonged mitosis (Fig. 8D,E). These results demonstrated that centrosomal

glutamylation is crucial for mitotic spindle formation and proper cell division.

## Discussion

We here describe a new method to precisely manipulate tubulin glutamylation at centrosomes/basal bodies. This was achieved by

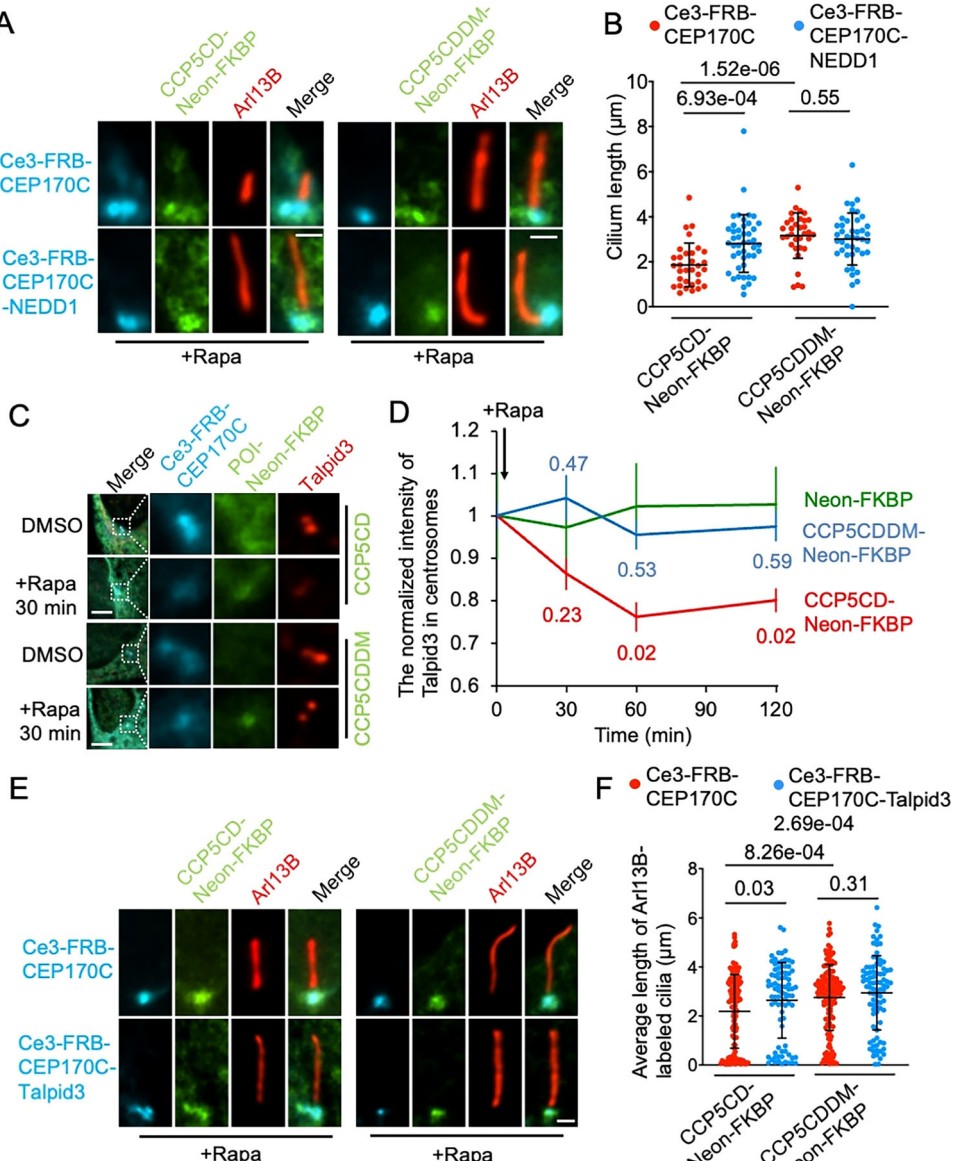

**Figure 7. Glutamylation regulates ciliogenesis via NEDD1 and Talpid3.**

(**A**) NIH3T3 cells co-transfected with the indicated constructs were serum-starved for 24 h and then treated with rapamycin (100 nM) for 60 min. Following treatment, cells were fixed and immunostained for Arl13B. Scale bar, 2 μm. (**B**) Quantification of cilium length of NIH3T3 cells from (**A**). Data (black) represent the mean ± SD. $n = 32$, 45, 33, and 42 cells from left to right, respectively. Three independent experiments. (**C**) NIH3T3 cells co-transfected with the indicated constructs were serum-starved for 24 h and then treated with rapamycin (100 nM) for the indicated times. Following treatment, cells were fixed and immunostained for Talpid3. Scale bar, 5 μm. (**D**) Normalized Talpid3 intensity in NIH3T3 cells from (**C**) after rapamycin treatment for the indicated times. Data represent the mean ± SEM. $n = 171$, 327 and 311 cells in the Neon-FKBP, CCP5CD-Neon-FKBP, and CCP5CDDM-Neon-FKBP groups, respectively. Three independent experiments. (**E**). NIH3T3 cells co-transfected with the indicated constructs were serum-starved for 24 h and then treated with rapamycin (100 nM) for 1 h. Following treatment, cells were fixed and immunostained for Arl13B. Scale bar, 1 μm. (**F**) Quantification of cilium length in NIH3T3 cells from (**E**). Data (black) represent the mean ± SD. $n = 134$, 83, 155, and 87 cells from left to right. Seven independent experiments. Student's $t$ tests were performed, and $P$ values are indicated. Source data are available online for this figure.

chemically recruiting an engineered deglutamylase, CCP5CD, to centrosomes. With spatial and temporal accuracy, we evaluated the causal relationships between glutamylation and centrosomes by monitoring the real-time effects of glutamylation reduction on centrosomes. Hypoglutamylation at centrosome has minimal impact on its structure (Fig. 2; Appendix Fig. S6A–F). In addition to structural considerations, glutamylation plays a critical role in

centrosome function (Appendix Fig. S12). Specifically, glutamylation physically recruits the NEDD1/CEP192/γ-tubulin complex through electrostatic forces to promote microtubule nucleation (Figs. 4 and EV2A–D; Appendix Fig. S8). The glutamylated centrosome-derived microtubules ensure proper centriolar satellite trafficking (Fig. 5). In G0 cells, glutamylation at basal bodies is vital for ciliogenesis and cilia maintenance (Figs. 6 and 7; Appendix

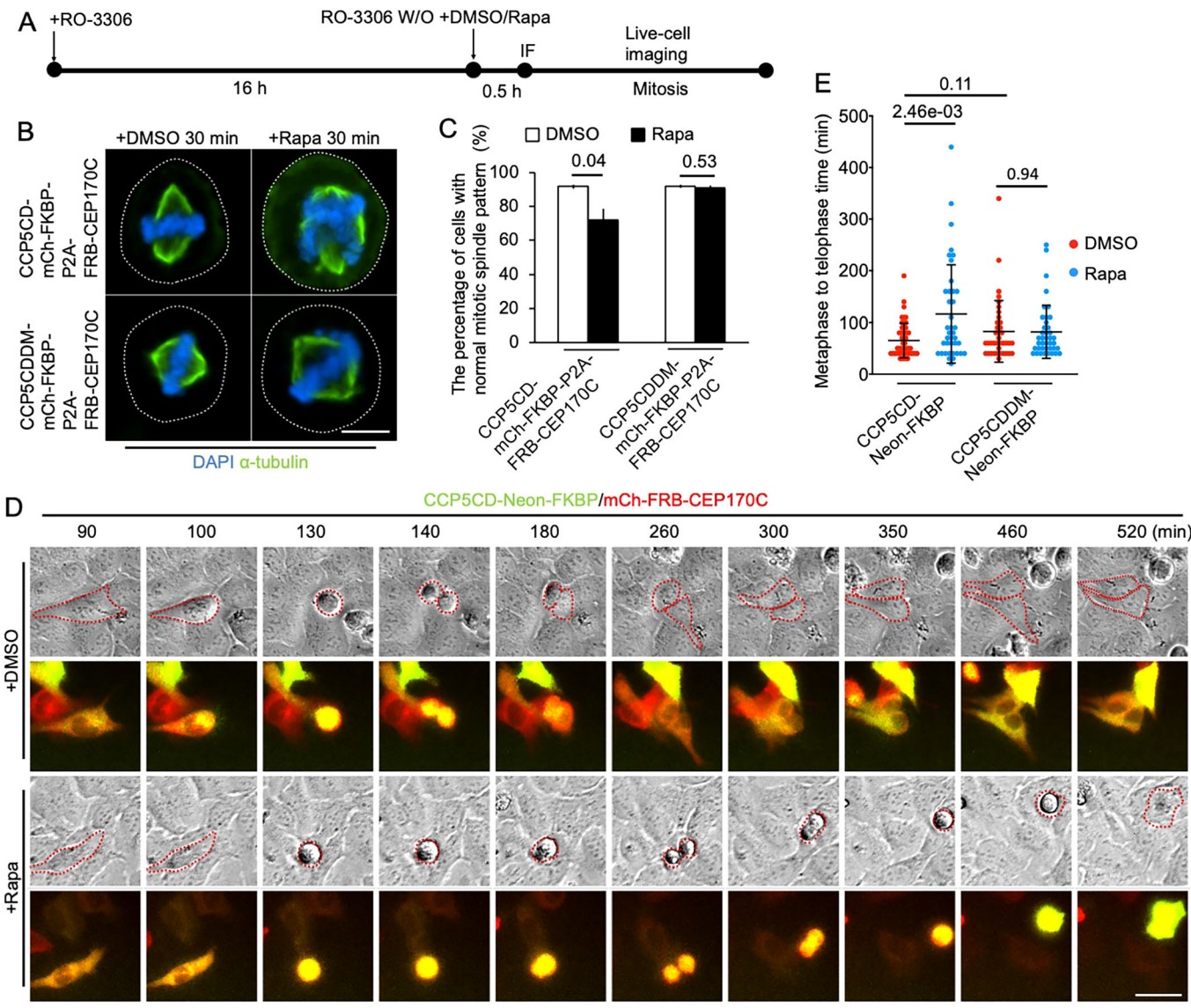

**Figure 8.   Centrosomal hypoglutamlyation perturbs mitotic spindle formation and prolongs mitosis.**

(A) Schematic of the protocol for RO-3306-induced cell synchronization at the G2/M transition, followed by rapamycin (Rapa)-induced centrosomal hypoglutamylation after RO-3306 washout (W/O). Cell morphology was analyzed by immunofluorescence (IF) 30 min after rapamycin or DMSO treatment. (B) HeLa cells co-transfected with the indicated constructs were synchronized at the G2/M transition and released into mitosis for 30 min in the presence of DMSO (0.1%) or rapamycin (Rapa, 100 nM). Mitotic spindles and chromosomes were labeled with anti-α-tubulin antibody (green) and DAPI (blue), respectively. Dashed lines indicate cell boundaries. Scale bar, 10 μm. (C) Quantification of the percentage of cells displaying a normal mitotic spindle pattern as shown in (B). Data are presented as mean ± SEM. $n = 90$ and 78 cells in CCP5CD and CCP5CDDM groups, respectively; three independent experiments. (D) Representative video frames of HeLa cells co-transfected with the indicated constructs, captured 30 min after treatment with DMSO (0.1%) or rapamycin (100 nM). Red dashed lines indicate the boundaries of mitotic cells. Scale bar, 10 μm. (E) Quantification of mitosis duration (from metaphase to telophase) in HeLa cells co-transfected with mCh-FRB-CEP170C and the indicated constructs, treated with DMSO (0.1%) or rapamycin (100 nM). Data are presented as mean ± SD (black). $n = 90$ and 78 cells in the CCP5CD and CCP5CDDM groups, respectively. Three independent experiments. Student's $t$ tests were performed, and $P$ values are indicated. Source data are available online for this figure.

Fig. S6I–K). NEDD1 and Talpid3 are involved in this process, likely due to functional centriolar satellites around glutamylated basal bodies (Appendix Fig. S12).

Early studies observed that long-lived microtubules, including those in centrioles and ciliary axonemes, undergo glutamylation (Janke, 2014; Audebert et al, 1993). This finding raised the possibility that glutamylation may regulate the intrinsic stability

of microtubules. However, acute reduction of glutamylation in ciliary axonemes by our CID system (Hong et al, 2018) and genetic approaches (He et al, 2018) does not affect the structural integrity of primary cilia. Moreover, we also found that centrosome structure remains largely unaffected by both short-term and long-term centrosomal hypoglutamylation (Fig. 2; Appendix Fig. S6A–F). Collectively, these results indicate that glutamylation is not

involved in the intrinsic stability of microtubules. Rather than intrinsic stability, glutamylation regulates the microtubule association of severing enzymes, which promotes microtubule disruption (Valenstein et al, 2016). Whether reducing glutamylation at centrioles modulates the affinity and activity of severing enzymes for microtubules remains an open question. A previous study masked glutamylated microtubules by injecting the GT335 antibody, which led to centriole disassembly (Bobinnec et al, 1998). Based on our findings, it is unlikely that centrosomal glutamylation inactivation directly causes centriole elimination. However, the impact of globally masking glutamylated tubulin on centrosomal structure warrants further investigation.

Our results reveal that glutamylation of centriolar tubulin physically recruits NEDD1 via electrostatic forces (Fig. 4K–R; Appendix Figs. S7 and S8). However, the cellular distribution of NEDD1 does not completely reflect the subcellular location of glutamylated microtubules (Haren et al, 2006). Glutamylated microtubules are abundant in ciliary axonemes, centrioles/basal bodies, mitotic spindles, and intercellular bridges (Janke and Magiera, 2020). Among these sites, NEDD1 mainly localizes to centrioles/basal bodies and to mitotic spindles, albeit at low density (Haren et al, 2006). Therefore, in addition to electrostatic forces, other mechanisms may be involved in determining NEDD1 distribution. For example, NEDD1 contains many potential phosphorylation sites (Lüders et al, 2006). During mitosis, Cdk1 may phosphorylate serine 411 of NEDD1, which increases its affinity for mitotic spindles (Lüders et al, 2006). A recent study found that PLK4-mediated phosphorylation of serine 325 in NEDD1 promotes its binding to nascent centrioles (Chi et al, 2021). Nonetheless, the details of the mechanism by which post-translational modifications of both NEDD1 and microtubules dictate NEDD1 distribution remain to be determined.

Mutation of the glutamylase TTLL5 causes cone photoreceptor dystrophy and infertility, likely owing to deficiencies of cilia and flagella (Bedoni et al, 2016). However, it is not clear why defects in TTLL5, which mainly localizes to centrosomes, lead to symptoms of ciliopathies. The method developed in the present study can spatiotemporally diminish glutamylation at centrosomes, allowing us to decipher the causative role of centrosomal glutamylation in cilia formation and maintenance. Aberrant centrosomal hypoglutamylation attenuates NEDD1/CEP192/r-tubulin-mediated microtubule nucleation. Such defects in centrosome-derived microtubules perturb centriolar satellite dynamics (Fig. 5), a pathway that regulates the levels of many key factors involved in ciliogenesis; they also suppress cilia formation and maintenance via Talpid3 signaling (Figs. 6 and 7). These pathways in TTLL5 mutant patients merit comprehensive scrutiny.

In summary, the methodology described here overcomes the challenge of genetically depleting multiple TTLL glutamylases for efficient deglutamylation. The inducible recruitment of an engineered deglutamylase to centrosomes endowed the spatial and temporal accuracy to manipulate centrosomal glutamylation level, allowing us to elucidate the causative mechanisms of how centrosomes regulate their structure and function via glutamylation. This tool offers a potential new strategy to treat glutamylation-related diseases.

# Methods

## Reagents and tools table

| Reagent/resource | Reference or source | Identifier or catalog number |
|---|---|---|
| **Experimental models** | | |
| HEK293T | ATCC | CRL-3216 |
| NIH3T3 | ATCC | CRL-1658 |
| U2OS | ATCC | HTB-96 |
| HeLa | ATCC | CCL-2 |
| **Recombinant DNA** | | |
| Cerulean3-FRB-CEP170C | This work | |
| CCP5CD-Neon-FKBP | This work | |
| CCP5CDDM-Neon-FKBP | This work | |
| GFP-CEP170FL | | |
| GFP-CEP170C | This work | |
| GFP-Centrin2 | This work | |
| GFP-Chibby | This work | |
| GFP-CPAP | This work | |
| GFP-Kizuna | This work | |
| GAIs-CFP-CEP170C | This work | |
| Neon-mGID1 | This work | |
| CCP5CD-mCherry-FKBP | This work | |
| EB1-YFP | This work | |
| Cerulean3-FRB-CEP170C-NEDD1 | This work | |
| Cerulean3-FRB-CEP170C-Neg-NEDD1 | This work | |
| PCM1F2-mCherry | This work | |
| Cerulean3-FRB-CEP170C-Talpid3 | This work | |
| mCherry-FRB-CEP170C | This work | |
| CCP5CD-mCherry-FKBP-P2A-FRB-CEP170C | This work | |
| CCP5CDDM-mCherry-FKBP-P2A-FRB-CEP170C | This work | |
| YFP-Centrin2 | This work | |
| mCherry-CEP170C | This work | |
| CCP5CD-mCherry-CEP170C | This work | |
| CCP5CDDM-mCherry-CEP170C | This work | |
| Neon-IFT88 | This work | |
| CEP192-Cerulean3-FRB-CEP170C | This work | |
| **Antibodies** | | |
| Mouse monoclonal anti-α-tubulin | Sigma-Aldrich | T6199 |
| Mouse monoclonal anti-polyglutamylation modification, (GT335) | Adipogen | AG-20B-0020-C100 |

| Reagent/resource | Reference or source | Identifier or catalog number |
|---|---|---|
| Mouse monoclonal anti-acetylated tubulin | Sigma-Aldrich | T7451 |
| Mouse monoclonal anti-Centrin2 | Merck | 04-1624 |
| Mouse polyclonal anti-γ-tubulin | Sigma-Aldrich | T6557 |
| Rabbit polyclonal anti-NEDD1 | Novus Biologicals | NBP1-83377 |
| Rabbit polyclonal anti-IFT88 | Proteintech | 13967-1-AP |
| Rabbit polyclonal anti-CP110 | Proteintech | 12780-1-AP |
| Rabbit polyclonal anti-ARL13B | Proteintech | 17711-1-AP |
| Rabbit polyclonal anti-tau | Dako | A0024 |
| Rabbit polyclonal anti-PCM1 | Proteintech | 19856-1-AP |
| Rabbit polyclonal anti-CEP290 | Abcam | ab84870 |
| Rabbit polyclonal anti-Pericentrin | Abcam | ab220784 |
| Rabbit polyclonal anti-Talpid3 | Proteintech | 24421-1-AP |
| Rabbit polyclonal anti-CEP164 | Proteintech | 22227-1-AP |
| Rabbit polyclonal anti-ODF2 | Sigma-Aldrich | HPA001874 |
| Rat monoclonal anti-mCherry | ThermoFisher | M11217 |
| Rabbit polyclonal anti-CEP83 | Gift from Dr. Wong-Jing Wang (National Yang Ming Chiao Tung University, Taipei, Taiwan) | N/A |
| Rabbit polyclonal anti-CEP192 | Proteintech | 18832-1-AP |
| Rabbit polyclonal anti-Polyglutamate chain (polyE) | AdipoGen | AG-25B-0030 |
| **Oligonucleotides and other sequence-based reagents** | | |
| ON-TARGETplus siRNAs | Horizon Discovery Ltd | Table EV1 |
| qPCR primer | This work | Table EV2 |
| **Chemicals, enzymes, and other reagents** | | |
| Rapamycin | LC Laboratories | CAS#53123-88-9 |
| RO-3306 | Sigma-Aldrich | SML0569 |
| Nocodazole | Sigma-Aldrich | M1404 |
| Paclitaxel | Sigma-Aldrich | T7402 |
| GA3-AM | Gift from Dr. Tasuku Ueno (University of Tokyo) | N/A |
| **Software** | | |
| Nikon Element AR software | Nikon | |
| Prism | GraphPad | |
| Zen | Zeiss | |

| Reagent/resource | Reference or source | Identifier or catalog number |
|---|---|---|
| Huygens deconvolution | Scientific Volume Imaging | |
| iBright™ FL1500 Instrument | Thermo Scientific | |

## Cell culture and transfection

NIH3T3, U2OS, HeLa, and HEK293 cells were maintained at 37 °C in 5% $CO_2$ in DMEM (Corning) supplemented with 10% fetal bovine serum (Gibco), penicillin, and streptomycin (Corning). NIH3T3 cells were transfected with plasmid DNA using the LT1 (Mirus) transfection reagent or X-tremeGENE9 (Roche), while U2OS and HeLa cells were transfected using Fugene (Promega), and HEK293 cells were transfected using LT1 (Mirus). To induce ciliogenesis, transfected NIH3T3 cells underwent serum starvation for 24 h. To induce rapid centrosomal hypoglutamylation, transfected cells were treated with 100 nM rapamycin (LC Laboratories) or 100 μM GA3-AM (Fan et al, 2017).

## Immunofluorescence staining

Cells were plated on poly(D-lysine)-coated borosilicate glass Lab-Tek eight-well chambers (Thermo Scientific). Cells were fixed in 100% methanol (Sigma-Aldrich) at –20 °C for 10 min (for labeling GT335, PolyE, α-tubulin, and centrosome proteins) or fixed with 4% paraformaldehyde (Electron Microscopy Sciences) at room temperature for 15 min (for labeling IFT88 and Arl13b labeling). Fixed cells were then permeabilized with 0.2% Triton X-100 (Sigma-Aldrich), followed by incubation in blocking solution (phosphate-buffered saline with 2% bovine serum albumin) for 30 min at room temperature. Primary antibodies were diluted in blocking solution, and samples were immersed in the diluted antibody mixtures for 1 h at room temperature. The primary antibodies used in this study against the following proteins: glutamylated tubulin (GT335 antibody, diluted 1:1000), polyglutamylated tubulin (PolyE antibody, diluted 1:1000), Arl13B (diluted 1:500), Talpid3 (diluted 1:500), IFT88 (diluted 1:500), PCM1 (diluted 1:500), Centrin2 (diluted 1:500), NEDD1 (diluted 1:500), Pericentrin (diluted 1:2000), α-tubulin (diluted 1:500), γ-tubulin (diluted 1:500), CEP164 (diluted 1:2000), ODF2 (diluted 1:500), CEP192 (diluted 1:200), and CEP83 (diluted 1:200). Secondary antibodies were diluted at 1:1000 in blocking solution, followed by incubation with samples for 1 h at room temperature.

## Epi-fluorescence imaging

Images of fixed cells were captured as multiple Z-stacks with a 0.5-μm thickness per stack using a Ti-E inverted fluorescence microscope (Nikon) with a ×40, ×60, or ×100 oil objective (Nikon) and a DS-Qi2 CMOS camera (Nikon). For EB1-positive microtubule tracks, COS7 cells were transfected with EB1-YFP and the indicated constructs. After 24 h of transfection, live-cell images were captured for 3 min at 5 s intervals with a Ti-E inverted fluorescence microscope with a 60× oil objective and DS-Qi2 CMOS camera at 37 °C in a 5% $CO_2$ atmosphere equipped with a

heating stage (Live Cell Instrument). For analysis of the distribution of centriolar satellites upon centrosomal hypoglutamylation, NIH3T3 cells were transfected with PCM1F2-mCh, and the indicated constructs. After 24 h of transfection, live-cell images were captured for 135 min at 1-min intervals with a Ti-E inverted fluorescence microscope as noted above. All images were processed with Huygens deconvolution (Scientific Volume Imaging). Image analysis was mainly conducted using Nikon Element AR software.

## 3D-SIM super-resolved imaging

Cells seeded on coverslips were incubated with blocking buffer (3% bovine serum albumin (wt/vol) and 0.1% Triton X-100 in phosphate-buffered saline) for 30 min at room temperature. Cells were then incubated with anti-acetylated-tubulin (1:40,000) and anti-mcherry (1:100), diluted in blocking buffer, for 2 h at room temperature. After incubation with primary antibodies, cells were incubated with Alexa Fluor 488- and Alexa Fluor 594-conjugated secondary antibodies (1:500; Molecular Probes) for 1 h at room temperature. Coverslips were mounted onto slides with mounting medium (DUO82040; Sigma). The 3D-SIM super-resolved images were acquired using an ELYRA PS.1 LSM-780 system (Carl Zeiss) equipped with a Plan Apochromat 63×/NA 1.4 oil-immersion objective. The raw images were reconstructed using ZEN software under default parameters.

## Microtubule regrowth assay

U2OS cells were plated on poly(D-lysine)-coated borosilicate glass Lab-Tek eight-well chambers (Thermo Scientific). After 24 h of transfection, cells were treated with 100 nM rapamycin or DMSO (0.1%) for 30 min. Transfected cells were incubated on ice for 40 min with nocodazole (3.3 µM) for 40 min to depolymerize microtubules and subsequently placed at room temperature for 3 min. The fixed samples were then immunostained with anti-α-tubulin (1:500 dilution).

## FRAP experiments

FRAP experiments were carried out using a Laser Scanning Confocal Microscope 780 (LSM-780, Zeiss) equipped with a ×63 oil-immersion objective (Zeiss). Before bleaching, three sequential images were taken to obtain a baseline of fluorescence intensity of Neon-IFT88. The centrosome region of cells expressing Neon-IFT88 was then photobleached and allowed to recover for 10 min, during which images were acquired every 30 s. The fluorescence intensity of Neon-IFT88 was analyzed using Nikon Element AR software.

## Purification of tubulin and microtubule co-sedimentation

Non-glutamylated tubulin was purified from non-transfected HEK293 cells, and glutamylated tubulin was purified from TTLL4C1-overexpressing HEK293 cells as previously described (Widlund et al, 2012). HEK293 cells were lysed in RIPA lysis buffer (50 mM Tris-HCl, 2 mM EGTA, 9% NaCl, 1% Triton X-100) containing protease inhibitors (Roche). The concentration of cell lysates was measured using the Bio-Rad Protein Assay. For microtubule co-sedimentation, 1 mg of HEK293 cell lysate was

added to non-glutamylated tubulin or glutamylated tubulin at a final concentration of 2 µM in the presence of 1 mM GTP and 20 µM paclitaxel, and then allowed to polymerize for 30 min at 37 °C. Samples were then centrifuged at 100,000×g for 30 min at room temperature. The microtubule pellet was resuspended in tubulin buffer (1 × BRB80, 10 µM [Mg$_2$Cl + GTP], 5% glycerol). The resuspended samples and 10% cell lysate (input) were diluted with 4 × Laemmli sample buffer and boiled at 95 °C for 10 min for subsequent Western blot analysis. Proteins were separated using SDS-PAGE and transferred to a polyvinylidene difluoride membrane. After transfer, the membrane was incubated in blocking buffer (Tris-buffered saline with 0.1% Tween-20 and 5% skim milk) for 1 h at room temperature and then stained for 1 h at room temperature with a primary antibody: anti-glutamylated tubulin GT335 (1:1000) or anti-NEDD1 (1:1000), each diluted in blocking buffer. Horseradish peroxidase-conjugated secondary antibodies were diluted in blocking buffer (anti-rabbit 1:10,000, anti-mouse 1:5000) and incubated with the membrane for 1 h at room temperature. Immunopositive bands were detected using Amersham™ ECL Select™ (GE Healthcare), and images were acquired with an iBright™ FL1500 Instrument (Thermo Scientific).

## RNA interference and qPCR

The knockdown efficiency of the commercial ON-TARGETplus siRNAs for mouse TTLL5 (Horizon Discovery Ltd., J-3669-05-08) in NIH3T3 cells was verified by quantifying mTTLL5 mRNA by qPCR. This experiment was designed according to the manufacturer's protocol, with the ON-TARGETplus GAPDH Control siRNA Pool for Mouse (Horizon Discovery Ltd., D-001830-20-05) as the positive control and the ON-TARGETplus Non-Targeting siRNA Pool for Mouse (Horizon Discovery Ltd., D-001801-10-20) as the negative control (see Table EV1).

NIH3T3 cells were pre-cultured either on a chamber slide or in a 6-well plate in antibiotic-free DMEM one day before transfection. To monitor transfection efficiency, the Venus (C1) plasmid was optionally co-transfected with the siRNAs in different groups using the DharmaFECT Duo Transfection Reagent (Horizon Discovery Ltd.). The final concentrations of the plasmid and the siRNAs were 1 µg/mL and 100 nM, respectively. Transfected cells were further incubated for 48 h before fixation or lysis.

For the examination of the siRNA knockdown efficiency, total mRNA from different groups was extracted with the EZ-RNA Total RNA Isolation kit (Biological Industries, 20-400-100) and subsequently subjected to full-length cDNA reverse transcription with MMLV reverse transcriptase (ProTech) and the oligo-dT primer (Yeastern Biotech, HYT013-A01). The relative amounts of mTTLL5 in the cDNA samples were then detected by qPCR using SYBR master mix (Bioline, BIO-98020) and the primers listed in Table EV2. The relative level of a housekeeping gene, namely β-actin or mouse GAPDH (mGAPDH), was used as the control for quantification.

## NEDD1 mutation

The predicted structure of human NEDD1, obtained from AlphaFold, shows the N-terminal region consisting of a circular solenoid domain with WD40 repeats. The exposed centrosome-binding domain is predominantly positively charged ($+9.562$) in a

cellular environment at pH 7.4. To identify conserved and non-conserved amino acids present in the centrosome-binding domain, the NEDD1 wild-type sequence was analyzed on ConSurf. The structure was then modified using UCSF Chimera, specifically targeting conserved domains to alter the net surface charge without disrupting the overall protein structure. Certain point mutations, including ARG6ASP, HIS35ASP, LYS58GLU, LYS65GLU, LYS79-GLU, LEU97GLU, LYS107GLU, SER140ASP, LYS206GLU, LYS229GLU, ARG278GLU, and LYS291GLU, were introduced using the built-in rotamer library, selecting each residue with the highest probability. Any clashes between neighboring residues were resolved through energy minimization, implemented using AMBER ff14SB force field with 100 steps of steepest descent. Superimposing both structures in PyMOL yielded a root-mean-square deviation (RMSD) value of 0.01 Å between alpha carbons, indicating minimal structural differences. Mutation sites visualization in Appendix Fig. S7B were performed using PyMOL.

## Statistical analysis

Statistical analysis was performed with an unpaired two-tailed Student's $t$ test, and an $F$ test was used to determine whether variances were equal. $P$ values of $<0.05$ were considered to reflect a statistically significant difference between values; $P \geq 0.05$ represents no significant difference, $P < 0.05$ represents a significant difference, and $P < 0.01$ represents a highly significant difference. To assess the differences among group in Fig. EV1A,B, a one-way analysis of variance (ANOVA) was conducted using GraphPad Prism 10.

## Data availability

This study does not contain data amendable to external repositories.

The source data of this paper are collected in the following database record: biostudies:S-SCDT-10_1038-S44318-025-00435-y.

## Peer review information

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

## Acknowledgements

This study was supported by the National Science and Technology Council (NSTC), Taiwan (NSTC grant numbers 113-2628-B-007-003 to YCL, 112-2311-B-007-005 to HCC, 112-2628-B-A49-009-MY3, 113-2320-B-A49-018-MY3 to WJW, and 114-2823-8-007 -003 to LHCW). We thank technical supports from the Next-Generation Nucleic Acid Drug (NGNAD) Platform for the National Core Facility for Biopharmaceuticals, NSTC, Taiwan.

## Author contributions

**Shi-Rong Hong**: Formal analysis; Validation; Investigation; Visualization; Methodology; Writing—original draft. **Yi-Chien Chuang**: Formal analysis; Investigation. **Wen-Ting Yang**: Conceptualization; Formal analysis; Validation; Investigation; Visualization; Writing—original draft. **Chiou-Shian Song**: Formal analysis; Investigation; Writing—original draft. **Hung-Wei Yeh**: Formal analysis; Validation; Investigation. **Bing-Huan Wu**: Formal analysis; Investigation. **I-Hsuan Lin**: Formal analysis; Investigation. **Po-Chun Chou**: Formal analysis; Investigation; Methodology. **Shiau-Chi Chen**: Formal analysis; Investigation. **Lohitaksh Sharma**: Formal analysis; Investigation; Methodology. **Jui-Chen Lu**: Formal analysis; Investigation. **Rou-Ying Li**: Formal analysis. **Ya-Chu Chang**: Formal analysis; Investigation. **Kuan-Ju Liao**: Investigation. **Hui-Chun Cheng**: Conceptualization; Supervision; Validation. **Won-Jing Wang**: Supervision; Funding acquisition. **Lily Hui-CHing Wang**: Resources; Supervision; Funding acquisition; Project administration. **Yu-Chun Lin**: Conceptualization; Formal analysis; Supervision; Funding acquisition; Validation; Investigation; Methodology; Writing—original draft; Project administration; Writing—review and editing.

Source data underlying figure panels in this paper may have individual authorship assigned. Where available, figure panel/source data authorship is listed in the following database record: biostudies:S-SCDT-10_1038-S44318-025-00435-y.

## Disclosure and competing interests statement

The authors declare no competing interests.

# Expanded View Figures

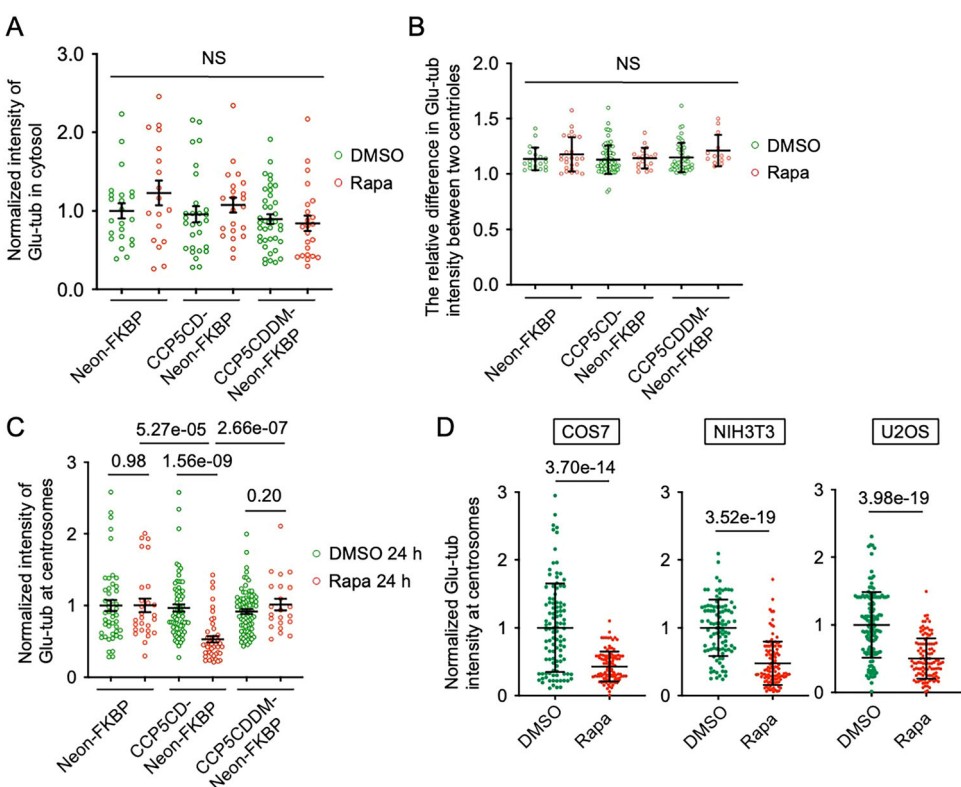

**Figure EV1. Specific glutamylation reduction occurs at two centrioles in three different cell types.**

(A–C) COS7 cells co-transfected with Ce3-FRB-CEP170C and the indicated constructs were treated with either 0.1% DMSO (green circles) or 100 nM rapamycin (Rapa, red circles) for 1 h in (A, B) or for 24 h in (C). Treated cells were immunostained with GT335 antibody to assess the level of glutamylated tubulin in cytosol (A); the relative difference in glutamylated tubulin intensity between two centrioles (B); and the normalized intensity of centrosomal glutamylation in (C). Data are presented as mean ± SEM. $n$ (from left to right) = 23, 18, 28, 22, 39, and 23 cells in (A); 21, 25, 78, 18, 46, and 13 cells in (B); 47, 27, 71, 46, 85, and 22 cells in (C), collected from 3 independent experiments. Statistical significance was determined using a one-way ANOVA test. "NS" indicates no significant difference among groups in (A, B). Student's t tests were performed, and P values are indicated in (C). (D) COS7, NIH3T3, and U2OS cells were transfected with Ce3-FRB-CEP170C and CCP5CD-Neon-FKBP. Transfected cells were treated with 0.1% DMSO (green) or rapamycin (100 nM) for 30 min. The normalized level of tubulin glutamylation at centrosomes under the indicated conditions is shown. Data (black) represent the mean ± SD. $n$ = 105, 107, 97, 113, 133, and 112 cells from left to right. Three independent experiments. Student's t tests were performed, and P values are indicated.

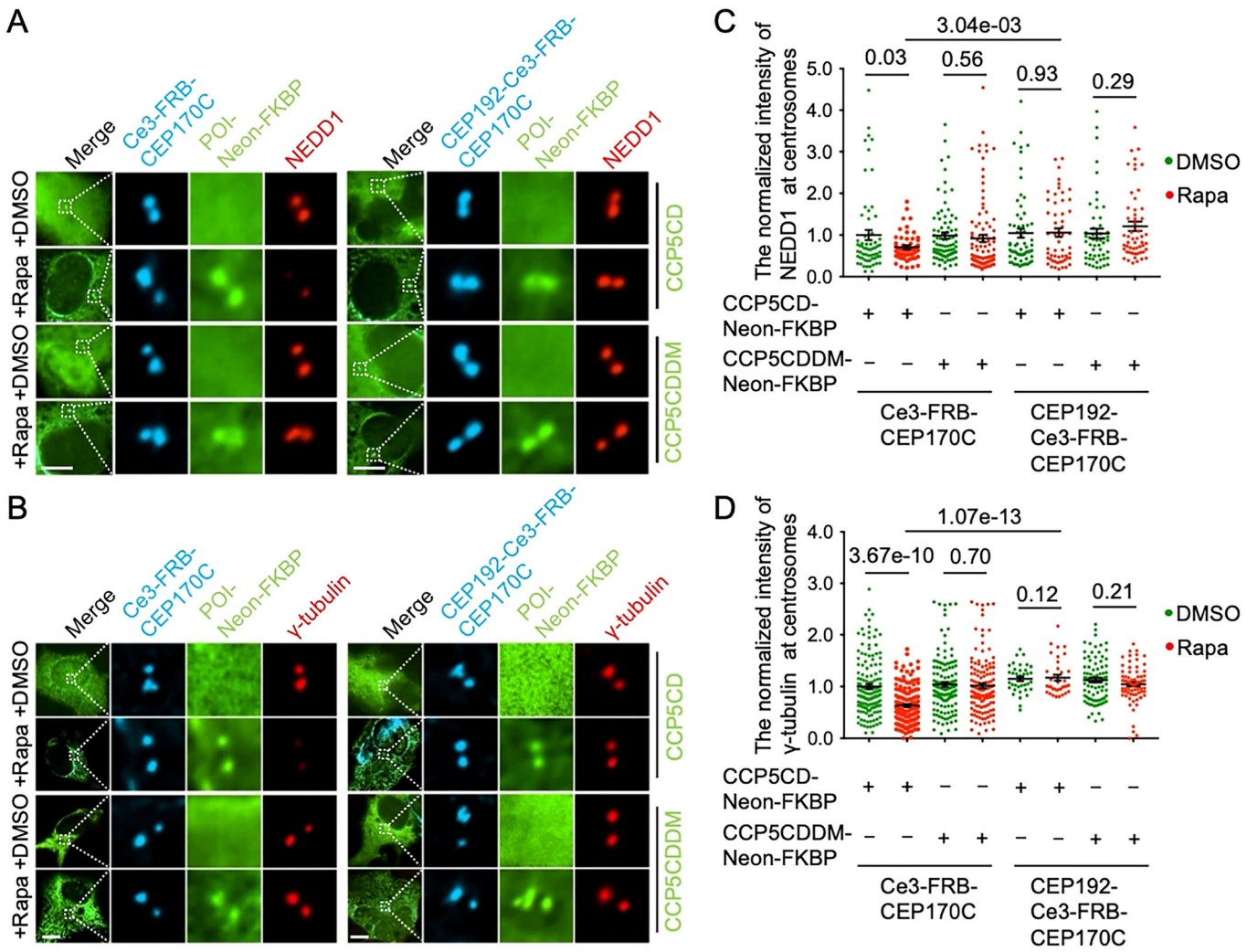

**Figure EV2.   CEP192 is sufficient to recruit NEDD1 and γ-tubulin to hypoglutamylated centrosomes.**

(A, B) COS7 cells co-transfected with the indicated constructs were treated with DMSO (0.1%) or Rapa (100 nM rapamycin) for 1 h. Following treatment, cells were immunostained with antibodies against NEDD1 (A) and γ-tubulin (B), respectively. Scale bar, 10 μm. (C, D) Quantification of the normalized intensity of NEDD1 (C) in cells from (A) and γ-tubulin (D) in cells from (B). Data represent as mean ± SEM. *n* (from left to right) = 62, 51, 84, 101, 67, 55, 53, and 54 cells in (C); 143, 152, 134, 135, 37, 36, 85, and 69 cells in (D); 3–6 independent experiments. Students' t tests were performed, and *P* values are indicated.

