## [Peer Review File · The EMBO Journal]

Glutamylation of Centrosomes Ensures their Function by Recruiting Microtubule Nucleation Factors

Shi-Rong Hong, Yi-Chien Chuang, Wen-Ting Yang, Chiou-Shian Song, Hung-Wei Yeh, Bing-Huan Wu, I-Hsuan Lin, Po-Chun Chou, Shiau-Chi Chen, Lohitaksh Sharma, Ruei-Zhen Lu, Rou-Ying Li, Ya-Chu Chang, Kuan-Ju Liao, Hui-Chun Cheng, Won-Jing Wang, Lily Wang, and Yu-Chun Lin

Corresponding author(s): Yu-Chun Lin (ycl@life.nthu.edu.tw)

Review Timeline:

Submission Date:	29th Jun 24
Editorial Decision:	28th Aug 24
Revision Received:	17th Feb 25
Editorial Decision:	26th Mar 25
Revision Received:	26th Mar 25
Accepted:	28th Mar 25

Editor: Hartmut Vodermaier

Transaction Report:

Dr. Yu-Chun Lin
National Tsing Hua University
Institute of Molecular Medicine, College of Life Science
101, Section 2, Kuang-Fu Road
Hsinchu City 300044
Taiwan

28th Aug 2024

Re: EMBOJ-2024-118291
Centrosomal Glutamylation Recruits the Microtubule Nucleation Factors to Ensure Its Functions

Dear Dr. Lin,

Thank you for submitting your study on the role of centrosomal glutamylation, and my sincere apologies for the delay in getting back to you with a decision - due to the summer vacation season, referee responses have been slower than usual. We have now received reports from three expert reviewers, copied below for your information. As you will see, all referees appreciate that the study addresses an important question, and that it reports unexpected and potentially interesting findings. At the same time, they raise a number of major concerns regarding the conclusiveness of various experiments at the present stage; including the key issue that the study only analyzes effects of acute rather than long-term deglutamylation, and that it has not demonstrated complete lack of centrosomal glutamylation. They also list several specific points related to controls and presentation.

Given the valuable new tool developed, and the potential overall interest, I would nevertheless remain open to considering a revised version further for EMBO Journal publication, in case that you should be able to decisively and comprehensively respond to the points listed in the three reports. However, since it is our policy to allow only a single round of (major) revision, it would be important that you contact me already during the early stages of your revision work with a revision plan based on a preliminary point-by-point response, so that we could discuss which experiments would be feasible and whether revision for The EMBO Journal would appear realistic. If necessary, we would also be open to extension of the default three-months revision period; our 'scooping protection' (meaning that competing work appearing elsewhere in the meantime will not affect our considerations of your study) would of course remain valid also throughout such an extension.

Detailed information on preparing, formatting and uploading a revised manuscript can be found below and in our Guide to Authors. Thank you again for the opportunity to consider this work for The EMBO Journal, and I look forward to hearing from you in due time.

Yours sincerely,

Hartmut Vodermaier

- size of the scale bars that are mandatory for all micrograph panels
- the statistical test used to generate error bars and P-values
- the type error bars (e.g., S.E.M., S.D.)
- the number (n) and nature (biological or technical replicate) of independent experiments underlying each data point
- Figures may not include error bars for experiments with $n < 3$; scatter plots showing individual data points should be used

instead.

9) To facilitate reproducibility and cross-laboratory adoption of methodologies, please structure the Materials & Methods section as outlined in our guide to authors, including a completed Reagents and Tools Table that can be downloaded from our author guidelines as well (<https://www.embopress.org/page/journal/14602075/authorguide#structuredmethods>).

10) Digital image enhancement is acceptable practice, as long as it accurately represents the original data and conforms to community standards. If a figure has been subjected to significant electronic manipulation, this must be clearly noted in the figure legend and/or the 'Materials and Methods' section. The editors reserve the right to request original versions of figures and the original images that were used to assemble the figure. Finally, we generally encourage uploading of numerical as well as gel/blot image source data; for details see: embopress.org/page/journal/14602075/authorguide#sourcedata

At EMBO Press, we ask authors to provide source data for the main manuscript figures. Our source data coordinator will contact you to discuss which figure panels we would need source data for and will also provide you with helpful tips on how to upload and organize the files.

Revision to The EMBO Journal should be submitted online within 90 days, unless an extension has been requested and approved by the editor; please click on the link below to submit the revision online before 26th Nov 2024:

Link Not Available

If you choose to alternatively have this study further considered by another EMBO Press publication, please use the following hyperlink to directly transfer the manuscript, optionally with inclusion of referee reports and identities:

Link Not Available

Referee #1:

Manuscript title: Centrosomal Glutamylation Recruits the Microtubule Nucleation Factors to Ensure Its Functions
Peer Review Summary and Evaluation

This study aims to delineate the functional roles of the microtubule glutamylation specifically on centriolar microtubules (MTs), a long-standing question in the field. Traditional methods in studying MT glutamylation rely on the manipulation of MT glutamylation occurring in multiple subcellular localization, thus precluding the functional analysis of MT glutamylation specifically at the centrosome. In this study, the authors overcame this technical problem by implementing the chemically inducible dimerization (CID)-based method. Upon acute rapamycin treatment, the engineered deglutamylase, CCP5CD-Neon-FKBP, dimerizes with the centrosome-localized peptide, Ce3-FRB-CEP170C, to performed localized deglutamylation of the

centriolar MT. Ce3-FRB-CEP170C expression did not affect centrosome structure and functions as evidenced by the staining of several centrosome/cilia markers, validating this CID-based technique as a reliable tool for specific functional analysis of centriolar glutamylation.

The acute deglutamylation of centriolar MTs disrupted centrosomal recruitment of MT nucleation factor NEDD1/CEP192/ γ -tubulin complex, leading to perturbed centrosomal MT nucleation and impaired MT-regulated functions including centriolar satellites (CS) distribution and cilia formation/maintenance. Intriguingly, centriolar deglutamylation did not affect several centrosome markers or centriole length. These data lead to the conclusion that centriole glutamylation is critical for centrosomal functions in MT nucleation but is dispensable for the structural integrity of centrosomes.

Mechanistically, expression of the engineered NEDD1 with constitutive centrosomal localization (Ce3-FRB-CEP170C-NEDD1) rescued the deglutamylation-induced phenotypes, including disrupted centrosomal γ -tubulin recruitment, reduced MT nucleation, altered CS distribution and impaired cilia formation, confirming that erroneous NEDD1 recruitment mediates the deglutamylation-induced phenotypes. The authors further showed that replacing the conserved positively charged residues to the negatively charged ones in NEDD1 perturbed its recruitment to the centrosome, suggesting that NEDD1 is recruited to the negatively charged glutamated side chain at the centriole.

Overall, this study provides convincing evidence to support the view that centriole glutamylation is required for accurate recruitment of MT-nucleating factors and MT-related functions. The CID-based method developed in this study is a reliable tool to investigate physiological roles of the site-specific MT glutamylation and has been nicely validated by appropriate control experiments. The finding that centriole glutamylation does not seem to contribute to the structural integrity of centrosomes is an unexpected novelty to the field. The manuscript is well-written and easy to follow. There are a few concerns that the authors should address before publication.

Major Points:

1. In Fig. S7, the authors argued in the corresponding text saying that long-term deglutamylation at the basal bodies (by CCP5CD-mch-CEP170C expression) resulted in shorter cilia compared with the inactive enzyme control (CCP5CDDM-mch-CEP170C). To me the inactive enzyme (CCP5CDDM-mch-CEP170C) is not a good control here because it did not rule out the possibility that such construct erroneously elongates cilia through some unexpected factors. The authors should incorporate mch-CEP170C construct alone as the control and compare cilia length between CCP5CD/CCP5CDDM-mch-CEP170C and mch-CEP170C to make this conclusion more convincing.
2. The study focused mainly on the short-term effects of the centriole deglutamylation, as most of the rapamycin treatment time ranging from 0.5-2 h. However, how the long-term centriole deglutamylation affects the structure of the centrosome is not addressed. Although CCP5CD-mCh-CEP170C-expressing cell line has been established as a tool to study the effects of long-term centriole deglutamylation on cilia formation (Fig. S7), it has not been applied to test centrosome structure. Since the traditional method of GT335 masking of the glutamylated MT was shown to disassemble centriole in 12 h (doi: 10.1042/BC20040112), it would be intriguing to see if long-term, centriole-specific deglutamylation causes similar phenotypes. The authors should examine a panel of centriole and PCM markers under extended rapamycin treatment in the CID system or in the CCP5CD-mCh-CEP170C-expressing cell line and compare the results to those obtained by GT335 masking method in the discussion.
3. The lack of effects in PCNT localization to the centrosome upon acute centriole deglutamylation is a bit surprising (Fig. 2C). PCNT is a well-known scaffold protein that organizes PCM architecture, and the protein adopts an elongating conformation with the C terminal "PACT" domain proximal to the centriole wall and the N-terminal extending outward into the PCM. Loss of PCNT also perturbed PCM accumulation and MT functions. The data presented in this study suggests that centriole glutamylation might not contribute to the binding of the PCNT PACT domain to the centriole wall. This might not be in the scope of the current study, but the authors should discuss possible PCNT-centriole wall binding mechanism if it is not via centriole glutamylation.

Minor Points:

1. In the summary, the sentence "Intriguingly, glutamylation physically recruited the NEDD1..." seems to have a redundant word "trafficking".
2. First sentence of the introduction seems to omit words "nucleating interphase MT and (establishing?) mitotic spindle"
3. There seems to be wrong labels in Fig. 4. Multiple Fig. 4E and F exists.
4. In several images, the pattern of the CCP5CD/CCP5CDDM-Neon-FKBP constructs recruited to centrosomes look quite blurry (Fig. 3C, Fig. 4A, C, E and Fig 5F), making it difficult to see if the constructs are properly recruited to the centrosome. They should be replaced with better quality images.
5. The 7th line of the first result: "We assumed that TLL5 does not solely regulate centrosomal glutamylation....." seems to be an unnecessary description for the whole sentence.
6. There seems to be a redundant sentence in the 3rd line of the second result.

Referee #2:

This manuscript entitled "Centrosomal Glutamylation Recruits the Microtubule Nucleation Factors to Ensure its Functions" by Hong, Yang, and Song et al describes the role of glutamylation modification in centrosome function. The study uses previously described engineered deglutamylase to localize to the centrosome and cell biology methods to dissect the effects of deglutamylation of centrosomal microtubules. The effects or functions of glutamylation are understudied especially in centrosome maintenance, therefore this manuscript attempts to understand a complex post-translation modification and may be of interest to cell biologists. Overall, the manuscript data is of good quality and well-written, I have a few concerns and

comments regarding the data presented here, its interpretation, and the model presented. In light of this, I recommend a revision to address the below comments before publication in its current form.

Some specific queries are outlined below;

1. The manuscript premises on the decreased levels of glutamylation of centrosomes either through their engineered CCP5CD-Neon-FKBP recruitment to centrosome or siRNA KO/KD of TLL5. However, the decrease in the levels of glutamylation shown in Figure S1C and Figure 1F is based on the immunofluorescence of GT335. It is important to demonstrate or quantify the levels of glutamylation either biochemically with western blot or through other antibodies such as PolyE.
2. While it is understood that the siRNA depletion of two or more TLLs might be tedious and time-consuming perhaps the CID method of deglutamylation is the way to go. The authors can check which are redundant TLLs express or overtake TLL5 KO/KD, which will be useful information for the community. Similar data can also be extracted after the addition of FKBP to the CCP5CD-Neon-FKBP/Ce3-FRB-CEP170C cells.
3. In Figure 1C and Figure 3C, upon the addition of FKBP the signal for Ce3-FRB-CEP170C is decreasing. Can the authors comment on this?
4. I have a hard time understanding the Figure 3 data. The authors should reconsider presenting the effects of microtubule nucleation or growth after centrosomal deglutamylation.
5. In the Figure 6, the authors show decreased cilia length. Does the axonemal microtubules show decreased glutamylation after the addition of FKBP to the CCP5CD-Neon-FKBP/Ce3-FRB-CEP170C?
6. The model presented in Figure 8 places NEDD1 as the key interactor between glutamylated microtubules and NEDD1/CEP192/gamma-tubulin complex. Can the authors confirm this using their Neg-NEDD1 construct used in Figure 4 M/N/O/P? This will strengthen their model and understanding of the functional significance of this interaction.
7. Minor points: The manuscript can avoid abbreviating MT throughout the text. The gamma-tubulin symbol is misplaced in the abstract.

Referee #3:

In this manuscript by Hong et al, the authors address a long-standing question regarding the role of tubulin glutamylation at centrosomes. To tackle this question, the authors used their elegant method to spatiotemporally deplete centrosomal tubulin glutamylation by recruiting an engineered deglutamylase and assessed the impact of acute depletion on centrosome functions. This is an important tool that will be very useful in the centrosome field. Interestingly, acute depletion does not seem to affect centrosome integrity but impacts the recruitment of pericentriolar components such as NEDD1/CEP192 and gamma tubulin and affects microtubule nucleation, satellite trafficking and ultimately ciliogenesis. Acute centrosomal deglutamylation has a clear impact on MT nucleation, a phenotype thought to be mediated by NEDD1, which binds glutamylated microtubules due to the electrostatic charges. Overall, the article is interesting but does not fully answer the role of glutamylation at the centrosomes but rather unveil the effect of acute hypo-glutamylation at centrosomes.

- specific major concerns essential to be addressed to support the conclusions

1 - While the authors show convincingly their approach of CID to target centrosomes, a couple of controls regarding the precise localization of CEP170C are missing.

GFP-CEP170C co-localization with GT335 is not resolutive enough using regular immunofluorescence. Using expansion microscopy, Rodriguez-Real et al (<https://doi.org/10.15252/embr.202256724>) reported that CEP170 localizes to subdistal appendages and below the centriole in the proximal region while it has been shown that GT335 covers the microtubule wall of centrioles (<https://doi.org/10.1038/s41592-020-0859-z>, <https://doi.org/10.1016/j.semcd.2021.12.001>, <https://doi.org/10.1002/cm.21870>), hinting for no colocalization. Could the authors comment this in their manuscript? In addition, can the authors test what is the precise localization of CEP170C by super resolution methods like expansion microscopy, does it mirror CEP170FL localization or is it evenly distributed at centrioles as they suggest?

Indeed, the authors mention that there is a co-localization at both centrioles (mother and daughter) but this would only correspond to the lower reported proximal signal seen by Rodriguez-Real et al and not the SDA signal (if existing for CEP170C). The choice of CEP170 may therefore explain why glutamylation depletion is only reduced by 50%. The remnant glutamylation signal questions some of the conclusions the authors have drawn, hence the suggestion of hypo-glutamylation rather than lack of glutamylation.

2- It would be interesting to couple CCP5 directly with CEP192 or NEDD1, which localizes to the entire surface of the centriole, close to the microtubule wall and test if the glutamylation depletion is more efficient.

3 - Figure 1E/F shows convincingly that CEP170C targets CCP5 deglutamylase to centrioles and this reduces by half the GT335 signal at centrosomes within 30 minutes. Can the authors quantify the signal at mother and daughter centrioles separately and test if there is a difference? This relates to the point that CEP170 localizes to SDA in mother centrioles and not daughter and if this is the same for CEP170C, there could be a difference in GT335 signal between the two (something that could be very interesting to test).

4- The authors demonstrate by various ways that centrioles integrity is not affected after 30 minutes of centrosomal deglutamylation (Figure 2). While this is convincing for the tested markers, it appears necessary to test longer depletion time as the glutamylation status might be important at certain stages of the centrosome function, for example during the G2/M transition when forces are applied to centrosomes. Also, the glutamylation status might be important during centriole assembly, something that cannot be tested with this short treatment. Can the authors test different longer timing?

5- The title "centrosome glutamylation is dispensable for the structural integrity of centrosome" is not supported by the results. The authors only looked at a few markers, and if the PCM is reduced, this means that the centrosome integrity might be affected. If the authors were thinking about centrioles, the remaining 50% of the signal may be protecting the centriole, so this title/conclusion is too strong. Here the structural integrity is not fully analyzed, super-resolution microscopy more resolutive than the SIM used, or a significant amount of electron microscopy imaging would be needed.

6 -Concerning the "negative NEDD1" protein, can the authors do the co-sedimentation assay to show that it cannot interact with glutamylated tubulin?

As these mutations are spread in the sequence, it is also possible that the non-centrosomal localization is due to protein misfolding and not to a lack of electrostatic interactions. This point needs to be tested, and numerical simulation does not appear to be sufficient.

NEDD1 is anchored to the centrosome by CEP192 and depletion of CEP192 leads to loss of NEDD1 at the centrosome (<https://doi.org/10.1038/s41467-021-26252-5>). A rescue with CEP192 relocated to the centrosome would be interesting to determine whether NEDD1 relocates and whether nucleation is restored.

7- Upon deglutamylation (Figure 6C), the cilium length is only slightly affected. Could the authors test after a longer SS? For example, what about after 24h? In addition, the authors are using an acute depletion here, so glutamylation may return to its normal level. Could the authors quantify the GT335 signal after 0h, 4h, and 24h?

It is also surprising to see in graph 6C bars at the 0-micron level, suggesting that unformed cilia are included in the quantification of cilia length, is it correct? Could the authors exclude those and redo the statistics as this can impact the results?

8- Although CCP5 is recruited to the centrosome, the enzyme is still active and also resides in the cytosol, probably affecting cytoplasmic microtubules. Did the authors look at the level of glutamylation in the cytosol? The effect on the satellites may come from a trafficking problem on the microtubules and not directly at the centrosome. This effect can be similar for the ciliogenesis. In general, the authors should test the level of glutamylation (GT335) more systematically in ciliogenesis.

9-The experiment in Figure S7 is very important but indicates that the phenotype on ciliogenesis is minimal. What about the number of cilia? the level of cytoplasmic glutamylation?

Other than ciliogenesis, what about a long-term depletion and the effects on the distribution of EB1, the formation of spindles in mitosis?

10 -Overall, the conclusions should be tone down, as the absence of glutamylation is not proven, it seems to rather be an hypo-glutamylation, so the headings in the sub-sections and the discussion should be adapted.

- minor concerns that should be addressed

-In the abstract, it is written γ -tubulin and not gamma tubulin.

- For the EB1-YFP experiments, the authors mention the "EB1 dynamics", here the dynamics are not tested, only the number of comets.

- as a comment: it has been shown by cryo-tomography that IFT trains interact above the basal body (transition zone: OI: [10.1126/science.abm6704](https://doi.org/10.1126/science.abm6704)), so the result obtained by the authors about the absence of phenotype on the IFT is logical.

-The Figure 2K with the 3D SIM figure shows that CCP5CD is located around the centrioles, more like PCM than at the centrioles themselves. The resolution makes it difficult to see the microtubule wall.

- The impact of NEDD1 after centrosome deglutamylation affects MT nucleation, which is probably the cause of defects in satellite trafficking and the resulting defects in ciliogenesis. Perhaps the authors could better discuss this point.

We would like to thank the reviewers and the editors for their constructive feedback and insightful suggestions. We have revised the manuscript accordingly and have provided a separate file with tracked changes. The manuscript has also been reformatted to comply with the EMBO journal guidelines. Significant changes are itemized below, followed by our point-by-point responses that address the reviewers' concerns (in blue).

Major changes we have made, including new items:

1. **Fig 3A, C. Centrosomal glutamylation is essential for microtubule nucleation.** Centrosome-derived microtubules and EB1-tracks are highlighted in Fig 3A (arrowheads) and Fig 3C, respectively.
2. **Fig 6B, C, D. Centrosomal glutamylation is required for ciliogenesis.** We evaluated the cilia length extending from hypoglutamylated basal bodies and the proportion of ciliated cells after 24 h of serum-starvation.
3. **Fig 8A-D. Centrosomal hypoglutamylation perturbs mitotic spindle formation and prolongs mitosis.** We examined the effects of centrosomal hypoglutamylation on mitotic spindle formation and mitosis duration.
4. **Fig EV1A-D. Specific glutamylation reduction occurs at two centrioles in three different cell types.**
5. **Fig EV2. CEP192 is sufficient to recruit NEDD1 and γ -tubulin to hypoglutamylated centrosomes.**
6. **Appendix Fig S2B. CEP170C localizes to glutamylated sites of centrosomes.** We evaluated the distribution of full length CEP170 (CEP170FL) and the C-terminal domain of CEP170 (CEP170C).
7. **Appendix Fig S5. Recruitment of CCP5CD to centrosomes does not affect the polyglutamylated tubulin levels.**
8. **Appendix Fig S6. Long-term centrosomal hypoglutamylation suppresses microtubule nucleation and cilia formation but has minimal impact on centrosome structure.**
9. **Appendix Fig S8. The charge of centrosome-binding domain in NEDD1 is not critical for γ -tubulin recruitment.**
10. **Appendix Fig S9. Hypoglutamylation at basal body reduces axonemal glutamylation.**
11. **Appendix Fig S12. Working model depicting how glutamylation at centrosomes/basal bodies regulate cellular architecture and activities.** We have added the role of centrosomal glutamylation in mitotic spindle formation and mitosis.

Referee #1:

Manuscript title: Centrosomal Glutamylation Recruits the Microtubule Nucleation Factors to Ensure Its Functions

Peer Review Summary and Evaluation

This study aims to delineate the functional roles of the microtubule glutamylation specifically on centriolar microtubules (MTs), a long-standing question in the field. Traditional methods in studying MT glutamylation rely on the manipulation of MT glutamylation occurring in multiple subcellular

localization, thus precluding the functional analysis of MT glutamylation specifically at the centrosome. In this study, the authors overcame this technical problem by implementing the chemically inducible dimerization (CID)-based method. Upon acute rapamycin treatment, the engineered deglutamylase, CCP5CD-Neon-FKBP, dimerizes with the centrosome-localized peptide, Ce3-FRB-CEP170C, to perform localized deglutamylation of the centriolar MT. Ce3-FRB-CEP170C expression did not affect centrosome structure and functions as evidenced by the staining of several centrosome/cilia markers, validating this CID-based technique as a reliable tool for specific functional analysis of centriolar glutamylation.

The acute deglutamylation of centriolar MTs disrupted centrosomal recruitment of MT nucleation factor NEDD1/CEP192/ γ -tubulin complex, leading to perturbed centrosomal MT nucleation and impaired MT-regulated functions including centriolar satellites (CS) distribution and cilia formation/maintenance. Intriguingly, centriolar deglutamylation did not affect several centrosome markers or centriole length. These data lead to the conclusion that centriole glutamylation is critical for centrosomal functions in MT nucleation but is dispensable for the structural integrity of centrosomes.

Mechanistically, expression of the engineered NEDD1 with constitutive centrosomal localization (Ce3-FRB-CEP170C-NEDD1) rescued the deglutamylation-induced phenotypes, including disrupted centrosomal γ -tubulin recruitment, reduced MT nucleation, altered CS distribution and impaired cilia formation, confirming that erroneous NEDD1 recruitment mediates the deglutamylation-induced phenotypes. The authors further showed that replacing the conserved positively charged residues to the negatively charged ones in NEDD1 perturbed its recruitment to the centrosome, suggesting that NEDD1 is recruited to the negatively charged glutamated side chain at the centriole.

Overall, this study provides convincing evidence to support the view that centriole glutamylation is required for accurate recruitment of MT-nucleating factors and MT-related functions. The CID-based method developed in this study is a reliable tool to investigate physiological roles of the site-specific MT glutamylation and has been nicely validated by appropriate control experiments. The finding that centriole glutamylation does not seem to contribute to the structural integrity of centrosomes is an unexpected novelty to the field. The manuscript is well-written and easy to follow. There are a few concerns that the authors should address before publication.

We appreciate the reviewer not only for recognizing the value of our approach but also for providing constructive expert comments.

Major Points:

1. In Fig. S7, the authors argued in the corresponding text saying that long-term deglutamylation at the basal bodies (by CCP5CD-mch-CEP170C expression) resulted in shorter cilia compared with the inactive enzyme control (CCP5CDDM-mch-CEP170C). To me the inactive enzyme (CCP5CDDM-mch-CEP170C) is not a good control here because it did not rule out the possibility that such construct erroneously elongates cilia through some unexpected factors. The authors should incorporate mch-CEP170C construct alone as the control and compare cilia length between CCP5CD/CCP5CDDM-mch-CEP170C and mch-CEP170C to make this conclusion more convincing.

We thank the reviewer for this constructive suggestion. We have measured the cilia length and proportion of ciliated cells in cells expressing mCh-CEP170C and included this control in the revised manuscript (Appendix Fig S6 I-K). As expected, the expression of CCP5CD-mCh-CEP170C, but not mCh-CEP170C or CCP5CDDM-mCh-CEP170C, significantly reduced both the cilia length and the proportion of ciliated cells.

Appendix Figure S6. Long-term hypoglutamylation at centrosomes suppresses cilia formation.

- I. NIH3T3 cells transfected with the indicated constructs were serum-starved for 24 h and then immunostained with Arl13 antibody (green) to visualize primary cilia. Scale bar, 10 μm.
- J,K. Quantification of cilia length (J) and proportion of ciliated cells (K) in the cells shown in (I). Data are presented as mean ± S.E.M. n (from left to right) = 72, 82, and 114 cells in (J); from 3-6 independent experiments. Students' *t*-tests were performed, and *P* values are indicated.

2. The study focused mainly on the short-term effects of the centriole deglutamylation, as most of the rapamycin treatment time ranging from 0.5-2 h. However, how the long-term centriole deglutamylation affects the structure of the centrosome is not addressed. Although CCP5CD-mCh-CEP170C-expressing cell line has been established as a tool to study the effects of long-term centriole deglutamylation on cilia formation (Fig. S7), it has not been applied to test centrosome structure. Since the traditional method of GT335 masking of the glutamylated MT was shown to disassemble centriole in 12 h (doi: 10.1042/BC20040112), it would be intriguing to see if long-term, centriole-specific deglutamylation causes similar phenotypes. The authors should examine a panel

of centriole and PCM markers under extended rapamycin treatment in the CID system or in the CCP5CD-mCh-CEP170C-expressing cell line and compare the results to those obtained by GT335 masking method in the discussion.

As the reviewer suggested, we examined the centrosome structure in cells expressing mCh-CEP170C, CCP5CD-mCh-CEP170C, and CCP5CDDM-mCh-CEP170C (Appendix Fig S6A-F). The centrioles, distal appendages, and pericentriolar matrix were labeled by Centrin2, CEP164, and Pericentrin, respectively. The expression of mCh-CEP170C, CCP5CD-mCh-CEP170C, or CCP5CDDM-mCh-CEP170C did not alter the pattern or intensity of these centrosomal markers, indicating that long-term hypoglutamylation at centrosomes has minimal impact on the centrosome structure.

We have added the content below to the discussion.

“Moreover, we also found that centrosome structure remains largely unaffected by both short-term and long-term centrosomal hypoglutamylation (Fig 2; Appendix Fig S6A-F). Collectively, these results indicate that glutamylation is not involved in the intrinsic stability of microtubules. Rather than intrinsic stability, glutamylation regulates the microtubule association of severing enzymes, which promotes microtubule disruption (Valenstein et al, 2016). Whether reducing glutamylation at centrioles modulates the affinity and activity of severing enzymes for microtubules remains an open question. A previous study masked glutamylated microtubules by injecting the GT335 antibody, which led to centriole disassembly (Bobinnec et al., J Cell Biol, 1998). Based on our findings, it is unlikely that centrosomal glutamylation inactivation directly causes centriole elimination. However, the impact of globally masking glutamylated tubulin on centrosomal structure warrants further investigation.”

Appendix Figure S6. Long-term centrosomal hypoglutamylation suppresses microtubule nucleation and cilia formation but has minimal impact on centrosome structure.

A,C,E. COS7 cells transfected with the indicated constructs were immunostained with the antibodies against Centrin2 (A), CEP164 (C), and Pericentrin (E), respectively. Lower panels show magnified images of the areas demarcated by the dashed squares. Scale bar, 10 μ m.

B,D,F. The normalized intensity of Centrin2, CEP164, and Pericentrin of cells in (A), (C), and (E), respectively. The solid red lines and dashed red lines in violin plots represent the median and first/third quartiles, respectively. n (from left to right) = 121, 122, and 116 cells in (B); 307, 210, and 115 cells in (D); 194, 248, and 194 cells in (F); 3~6 independent experiments.

Students' *t*-tests were performed, and *P* values are indicated.

3. The lack of effects in PCNT localization to the centrosome upon acute centriole deglutamylation is a bit surprising (Fig. 2C). PCNT is a well-known scaffold protein that organizes PCM architecture, and the protein adopts an elongating conformation with the C terminal "PACT" domain proximal to the centriole wall and the N-terminal extending outward into the PCM. Loss of PCNT also perturbed PCM accumulation and MT functions. The data presented in this study suggests that centriole glutamylation might not contribute to the binding of the PCNT PACT domain to the centriole wall.

This might not be in the scope of the current study, but the authors should discuss possible PCNT-centriole wall binding mechanism if it is not via centriole glutamylation.

We thank the reviewer for the constructive suggestion and have added the following content to the revised manuscript.

“Pericentrin is a known scaffold protein that organizes pericentriolar architecture and facilitates microtubule nucleation (Delaval & Doxsey, 2010). The minimal impact of centrosomal hypoglutamylation on pericentrin suggests that glutamylation regulates microtubule nucleation through pericentrin-independent mechanisms (Fig 2C and D; Appendix Fig S6G and H).”

Reference:

Delaval B, Doxsey SJ. Pericentrin in cellular function and disease. *J Cell Biol.* 2010 Jan 25;188(2):181-90.

Minor Points:

1. In the summary, the sentence "Intriguingly, glutamylation physically recruited the NEDD1..." seems to have a redundant word "trafficking".
2. First sentence of the introduction seems to omit words "nucleating interphase MT and (establishing?) mitotic spindle"
3. There seems to be wrong labels in Fig. 4. Multiple Fig. 4E and F exists.
4. In several images, the pattern of the CCP5CD/CCP5CDDM-Neon-FKBP constructs recruited to centrosomes look quite blurry (Fig. 3C, Fig. 4A, C, E and Fig 5F), making it difficult to see if the constructs are properly recruited to the centrosome. They should be replaced with better quality images.
5. The 7th line of the first result: "We assumed that TLL5 does not solely regulate centrosomal glutamylation....." seems to be an unnecessary description for the whole sentence.
6. There seems to be a redundant sentence in the 3rd line of the second result.

We thank the reviewer for bringing these to our attention and have corrected them accordingly.

Referee #2:

This manuscript entitled "Centrosomal Glutamylation Recruits the Microtubule Nucleation Factors to Ensure its Functions" by Hong, Yang, and Song et al describes the role of glutamylation modification in centrosome function. The study uses previously described engineered deglutamylase to localize to the centrosome and cell biology methods to dissect the effects of deglutamylation of centrosomal microtubules. The effects or functions of glutamylation are understudied especially in centrosome maintenance, therefore this manuscript attempts to understand a complex post-translation modification and may be of interest to cell biologists. Overall, the manuscript data is of good quality and well-written, I have a few concerns and comments regarding the data presented here, its interpretation, and the model presented. In light of this, I recommend a revision to address the below comments before publication in its current form.

We thank the reviewer for the positive comments and have addressed the questions accordingly in the revised manuscript.

Some specific queries are outlined below;

1. The manuscript premises on the decreased levels of glutamylation of centrosomes either through their engineered CCP5CD-Neon-FKBP recruitment to centrosome or siRNA KO/KD of TLL5. However, the decrease in the levels of glutamylation shown in Figure S1C and Figure 1F is based on the immunofluorescence of GT335. It is important to demonstrate or quantify the levels of glutamylation either biochemically with western blot or through other antibodies such as PolyE.

We thank the reviewer for suggestions. Glutamylation occurs not only at the centrosomes but also in other subcellular regions, including primary cilia, intercellular bridges, and mitotic spindles (Janke and Magiera, *Nat Rev Mol Cell Biol*, 2020). Since western blotting measures the total cellular glutamylation level, it does not accurately reflect the extent of centrosome-specific hypoglutamylation.

As the reviewer suggested, we have measured the level of polyglutamylated tubulin after recruiting CCP5CD-Neon-FKBP to centrosomes by immunostaining with the PolyE antibody. GT335 and PolyE antibodies specifically recognizes branch point glutamate and polyglutamylated tubulin with >3 glutamate residues, respectively (Magiera and Janke *Methods Cell Biol.* 2013). Recruitment of CCP5CD-Neon-FKBP to centrosomes specifically reduced the GT335-labeled glutamylated tubulin, with minimal effect on polyglutamylated tubulin labeled by PolyE antibody (Appendix Fig S5). Wang et al have demonstrated that overexpression of CCP5 only reduces GT335-labeled tubulin but not PolyE-labeled tubulin (Wang *et al.*, *BMC Biology*, 2023). Moreover, our previous study also showed that recruitment of CCP5CD-Neon-FKBP to primary cilia specifically reduces GT335-labeled tubulin but did not affect PolyE-labeled tubulin in cilia (Hong *et al.*, *Nat Commun.*, 2018). Taken together, we demonstrated that CCP5 preferentially depletes glutamylated tubulin while having minimal impact on long-chain polyglutamated tubulin.

We have added the content and results to the revised manuscript.

“Consistent to previous studies (Hong *et al.*, *Nat Commun.*, 2018; Wang *et al.*, *BMC Biol.*, 2023). CCP5CD preferentially reduced GT335-labeled glutamylated tubulin while having minimal impact on long-chain polyglutamated tubulin detected by PolyE antibody (Appendix Fig S5).”

Appendix Figure S5. Recruitment of CCP5CD to centrosomes does not affect the polyglutamylated tubulin levels.

- A. COS7 cells were co-transfected with Ce3-FRB-CEP170C (green) and the indicated constructs, followed by treatment with either 0.1% DMSO or 100 nM rapamycin (Rapa) for 1 h. Cells were then immunostained with a PolyE-tubulin antibody (red). Scale bar, 10 μ m.
- B. Quantification of normalized PolyE tubulin intensity at centrosomes of cells from (A). Data are presented as mean \pm SD (green: DMSO; red: rapamycin). n = 59, 22, 45, 30, 39, and 45 cells from left to right; 3 independent experiments. Students' *t*-tests were performed, and *P* values are indicated.

Reference:

Magiera MM, Janke C. Investigating tubulin posttranslational modifications with specific antibodies. *Methods Cell Biol.* 2013;115:247-67.

Wang Y, Zhang Y, Guo X, Zheng Y, Zhang X, Feng S, Wu HY. CCP5 and CCP6 retain CP110 and negatively regulate ciliogenesis. *BMC Biol.* 2023 May 24;21(1):124.

Hong SR, Wang CL, Huang YS, Chang YC, Chang YC, Pusapati GV, Lin CY, Hsu N, Cheng HC, Chiang YC, Huang WE, Shaner NC, Rohatgi R, Inoue T, Lin YC. Spatiotemporal manipulation of ciliary glutamylation reveals its roles in intraciliary trafficking and Hedgehog signaling. *Nat Commun.* 2018 Apr 30;9(1):1732.

2. While it is understood that the siRNA depletion of two or more TTLs might be tedious and time-consuming perhaps the CID method of deglutamylation is the way to go. The authors can check which are redundant TTLs express or overtake TTL5 KO/KD, which will be useful information for

the community. Similar data can also be extracted after the addition of FKBP to the CCP5CD-Neon-FKBP/Ce3-FRB-CEP170C cells.

We agree with the reviewer that identifying key enzymes for centrosomal glutamylation is important. Almost all TLL members localize to centrosomes, implying their potential roles in centrosomal glutamylation (including TLL1, 4, 5, 6, 7, 9, and 11; van Dijk et al., *Mol Cell*. 2007). Moreover, many TLLs are preferentially abundant in different tissues (Fullston et al., *Am J Med Genet B Neuropsychiatr Genet*. 2011; Bosch Grau et al., *J Cell Biol*, 2013; Ikegami et al., *J Biol Chem*, 2006; Wu et al., *PLoS Genet*. 2022). The key glutamylases identified in NIH3T3 cells or COS7 cells (if any) may not be the same in other tissues. Since this is beyond the scope of our current research, we preferred to further emphasize the difficulty of deglutamylation using TLL depletion instead of experimentally exploring the key TLLs for centrosomal glutamylation.

We have added the sentences below to the revised manuscript.

“Moreover, many TLLs are preferentially abundant in different tissues (Fullston et al., *Am J Med Genet B Neuropsychiatr Genet*. 2011; Bosch Grau et al., *J Cell Biol*, 2013; Ikegami et al., *J Biol Chem*, 2006; Wu et al., *PLoS Genet*. 2022), making it challenging to use TLL knockdown to reduce centrosomal glutamylation in distinct systems.”

Reference:

van Dijk J, Rogowski K, Miro J, Lacroix B, Eddé B, Janke C. A targeted multienzyme mechanism for selective microtubule polyglutamylation. *Mol Cell*. 2007 May 11;26(3):437-48.

Fullston T, Gabb B, Callen D, Ullmann R, Woollatt E, Bain S, Ropers HH, Cooper M, Chandler D, Carter K, Jablensky A, Kalaydjieva L, Gecz J. Inherited balanced translocation t(9;17)(q33.2;q25.3) concomitant with a 16p13.1 duplication in a patient with schizophrenia. *Am J Med Genet B Neuropsychiatr Genet*. 2011 Mar;156(2):204-14.

Bosch Grau M, Gonzalez Curto G, Rocha C, Magiera MM, Marques Sousa P, Giordano T, Spassky N, Janke C. Tubulin glycolases and glutamylases have distinct functions in stabilization and motility of ependymal cilia. *J Cell Biol*. 2013 Aug 5;202(3):441-51.

Ikegami K, Mukai M, Tsuchida J, Heier RL, Macgregor GR, Setou M. TLL7 is a mammalian beta-tubulin polyglutamylation required for growth of MAP2-positive neurites. *J Biol Chem*. 2006 Oct 13;281(41):30707-16.

Wu HY, Rong Y, Bansal PK, Wei P, Guo H, Morgan JI. TLL1 and TLL4 polyglutamylases are required for the neurodegenerative phenotypes in *pcd* mice. *PLoS Genet*. 2022 Apr 11;18(4):e1010144.

3. In Figure 1C and Figure 3C, upon the addition of FKBP the signal for Ce3-FRB-CEP170C is decreasing. Can the authors comment on this?

We thank the reviewer for highlighting this point. In Fig 1C, the signal decline of Ce3-FRB-CEP170C following its dimerization with Neon-FKBP is due to FRET (Fluorescence Resonance Energy Transfer), where the emission light of Cerulean3 is transferred to Neon as the two fluorescent proteins come into close proximity. In Fig 3C, the slight decrease in Ce3-FRB-CEP170C is likely attributable to photobleaching caused by the frequent imaging of EB-1 dynamics.

4. I have a hard time understanding the Figure 3 data. The authors should reconsider presenting the effects of microtubule nucleation or growth after centrosomal deglutamylation.

To address the reviewer's suggestion, we have added hollow arrowheads in Fig 3A to indicate the centrosome-derived microtubules (α -tubulin-labeled filaments extending from the centrosomal region). Moreover, the EB1-tracks originating from centrosomes were depicted in Fig 3C.

Figure 3. Centrosomal glutamylation is essential for microtubule nucleation

- A. COS7 cells were co-transfected with Ce3-FRB-CEP170C and either CCP5CD-Neon-FKBP or CCP5CDDM-Neon-FKBP. At 80-90% confluency, transfected cells were treated with 100 nM rapamycin (Rapa) or 0.1% DMSO for 30 min and then placed on ice for 40 min to depolymerize microtubules. Cells were subsequently washed with DMEM for 1 min, allowed to recover for 3 min at room temperature (RT), and then fixed for immunostaining with anti- α -tubulin antibody. The right panels show magnified images of the areas outlined by the dashed squares. Hollow arrowheads indicate regrown microtubules, as shown by α -tubulin-labeled filaments extending from the centrosomal region. Scale bar, 10 μ m.
- B. Quantification of centrosome-derived microtubule intensity with or without 3 min recovery in cells transfected with the indicated constructs. Data represent the mean \pm S.E.M. (n = 270, 499, and 439

cells for the Neon-FKBP (green), CCP5CD-Neon-FKBP (red), and CCP5CDDM-Neon-FKBP groups (blue), respectively, from four to five independent experiments).

- C. COS7 cells were co-transfected with Ce3-FRB-CEP170C, CCP5CD-mCh-FKBP, and EB1-YFP. One day post-transfection, EB1-YFP-positive comets were monitored by live-cell imaging in the same cell before and after rapamycin-induced centrosomal hypoglutamylation. Centrosomal microtubule tracks were drawn based on EB1-YFP time-lapse imaging with 2 min duration. Insets show higher-magnification images of the centrosomal regions. Scale bar, 10 μ m.
- D. Quantification of EB1-YFP comet frequency emitted from centrosomes in cells co-transfected with Ce3-FRB-CEP170C and the indicated constructs before and after treatment with rapamycin (100 nM) for 30 min. $n = 11, 13$ and 10 cells in the Neon-FKBP, CCP5CD-Neon-FKBP and CCP5CDDM-Neon-FKBP groups, respectively, from four to five independent experiments.

Student's t -tests were performed, and P values are indicated.

5. In the Figure 6, the authors show decreased cilia length. Does the axonemal microtubules show decreased glutamylation after the addition of FKBP to the CCP5CD-Neon-FKBP/Ce3-FRB-CEP170C?

As suggested by the reviewer, we measured the level of axonemal glutamylation after recruiting Neon-FKBP, CCP5CD-Neon-FKBP, or CCP5CDDM-Neon-FKBP to CEP170C-labeled centrosomes. After 24 h of serum starvation, axonemal glutamylation in cilia extending from hypoglutamylated basal bodies was lower than in the control groups (Appendix Fig S9). However, since relative axoneme glutamylation (axonemal glutamylation/cilia length) remained comparable between glutamylated and hypoglutamylated basal bodies, we inferred that the lower axonemal glutamylation level resulted from the shorter cilia growing from hypoglutamylated basal bodies.

Appendix Figure S9. Hypoglutamylation at basal bodies reduces axonemal glutamylation.

- A. NIH3T3 cells co-transfected with Ce3-FRB-CEP170C (blue) and CCP5CD-Neon-FKBP (green) were pretreated with 100 nM rapamycin for 5 min. The treated cells were then serum-starved for 24 h and immunostained with the GT335 antibody (red). Scale bar, 10 μ m.

B. Quantification of normalized glutamylated tubulin intensity at ciliary axonemes. Data represent the mean \pm S.E.M. $n = 18, 22, 18, 23, 12,$ and 10 cells from left to right; 3 independent experiment. Students' t -tests were performed, and P values are indicated.

6. The model presented in Figure 8 places NEDD1 as the key interactor between glutamylated microtubules and NEDD1/CEP192/gamma-tubulin complex. Can the authors confirm this using their Neg-NEDD1 construct used in Figure 4 M/N/O/P? This will strengthen their model and understanding of the functional significance of this interaction.

To address the reviewer's suggestion, we tested whether negatively charged NEDD1 (Neg-NEDD1) at centrosomes can recruit γ -tubulin. Neg-NEDD1 was targeted to the centrosomes by tagging it with CEP170C, and the resulting fusion protein (Ce3-FRB-CEP170C-Neg-NEDD1) was still able to recruit γ -tubulin to centrosomes (Appendix Fig S8), suggesting that the charge of centrosome-binding domain in NEDD1 is not critical for γ -tubulin recruitment.

Haren et al. demonstrated that the N-terminus and C-terminus of NEDD1 protein are responsible for distinct functions: centrosome binding and γ -tubulin interaction, respectively (Haren et al, 2006). In line with this, our findings suggest that introducing negatively charged mutations into the N-terminus of NEDD1 affects its centrosome binding but does not impact its interaction with γ -tubulin.

Appendix Figure S8. The charge of centrosome-binding domain in NEDD1 is not critical for γ -tubulin recruitment.

NIH3T3 cells were co-transfected with Ce3-FRB-CEP170C and CCP5CD-Neon-FKBP or with Ce3-FRB-CEP170C-neg-NEDD1 (a negatively charged NEDD1 mutant) and CCP5CDDM-Neon-FKBP. Transfected cells were incubated with 100 nM rapamycin (Rapa) for 1 h and then immunostained for γ -tubulin. The normalized intensity of γ -tubulin under the indicated conditions is presented as mean \pm S.E.M. $n = 37, 41, 61,$ and 52 cells from left to right; 3 independent experiments. Student's t -tests were performed, and P values are indicated.

Reference:

Haren L, Remy MH, Bazin I, Callebaut I, Wright M, Merdes A. NEDD1-dependent recruitment of the gamma-tubulin ring complex to the centrosome is necessary for centriole duplication and spindle assembly. *J Cell Biol.* 2006 Feb 13;172(4):505-15.

7. Minor points: The manuscript can avoid abbreviating MT throughout the text. The gamma-tubulin symbol is misplaced in the abstract.

We have replaced “MT” with “microtubule” according and have corrected the misplaced symbol.

Referee #3:

In this manuscript by Hong et al, the authors address a long-standing question regarding the role of tubulin glutamylation at centrosomes. To tackle this question, the authors used their elegant method to spatiotemporally deplete centrosomal tubulin glutamylation by recruiting an engineered deglutamylase and assessed the impact of acute depletion on centrosome functions. This is an important tool that will be very useful in the centrosome field. Interestingly, acute depletion does not seem to affect centrosome integrity but impacts the recruitment of pericentriolar components such as NEDD1/CEP192 and gamma tubulin and affects microtubule nucleation, satellite trafficking and ultimately ciliogenesis. Acute centrosomal deglutamylation has a clear impact on MT nucleation, a phenotype thought to be mediated by NEDD1, which binds glutamylated microtubules due to the electrostatic charges. Overall, the article is interesting but does not fully answer the role of glutamylation at the centrosomes but rather unveil the effect of acute hypo-glutamylation at centrosomes.

We appreciate the positive comments from the reviewer.

- specific major concerns essential to be addressed to support the conclusions

1 - While the authors show convincingly their approach of CID to target centrosomes, a couple of controls regarding the precise localization of CEP170C are missing.

GFP-CEP170C co-localization with GT335 is not resolutive enough using regular immunofluorescence. Using expansion microscopy, Rodriguez-Real et al

(<https://doi.org/10.15252/embr.202256724>) reported that CEP170 localizes to subdistal appendages and below the centriole in the proximal region while it has been shown that GT335 covers the microtubule wall of centrioles ([https://doi.org/10.1038/s41592-020-0859-](https://doi.org/10.1038/s41592-020-0859-z)

[z, https://doi.org/10.1016/j.semcd.2021.12.001](https://doi.org/10.1016/j.semcd.2021.12.001), <https://doi.org/10.1002/cm.21870>), hinting for no colocalization. Could the authors comment this in their manuscript? In addition, can the authors test what is the precise localization of CEP170C by super resolution methods like expansion microscopy, does it mirror CEP170FL localization or is it evenly distributed at centrioles as they suggest?

Indeed, the authors mention that there is a co-localization at both centrioles (mother and daughter) but this would only correspond to the lower reported proximal signal seen by Rodriguez-Real et al and not the SDA signal (if existing for CEP170C). The choice of CEP170 may therefore explain why glutamylation depletion is only reduced by 50%. The remnant glutamylation signal questions some of

the conclusions the authors have drawn, hence the suggestion of hypo-glutamylated rather than lack of glutamylation.

We fully agree with the reviewer that CEP170FL specifically localizes to the mother centriole. Consistent with previous studies (Ma et al., eLife, 2022; Rodríguez-Real et al., EMBO Reports, 2023), GFP-tagged full-length CEP170 (GFP-CEP170FL) overlaps with mother centriole marker CEP164 (Appendix Fig S2A and B). In contrast, the C-terminus of CEP170 (CEP170C) is distributed to both the mother and daughter centrioles, as evidenced by its colocalization with various mother and daughter centriole markers (Appendix Fig S2A and B).

Although CEP170C is the strongest candidate for colocalization with centrosomal glutamylated sites in our tests (Appendix Fig S2C), it only partially overlaps with centrosomal glutamylated tubulin (Appendix Fig S2B and D). This may explain why the recruitment of CCP5CD to CEP170C-labeled sites does not completely deplete glutamylated tubulin at centrosomes.

As the reviewer suggested, we have replaced “hypoglutamylated” with “deglutamylated” in the revised manuscript.

Appendix Figure S2. CEP170C localizes to glutamylated sites of centrosomes

- A. Diagram of full-length CEP170 (CEP170FL; residues 1–1460) and its C-terminal portion (CEP170C; residues 1015–1460). The black and gray boxes represent the forkhead-associated domain and serine-rich domain, respectively.
- B. NIH3T3 cells transfected with GFP-CEP170FL or GFP-CEP170C (green) were immunostained with antibody against CEP164, Pericentrin, ODF2, SAS6, and Centrobin (red), respectively. The right panels show magnified images of the areas outlined by the dashed squares. Scale bar, 10 μ m.
- C. 3D-SIM image of NIH3T3 cells transfected with GFP-CEP170C (green) and immunostained with antibody GT335 (red). The right panels show magnified images of the areas outlined by the dashed squares. Scale bar, 10 μ m.
- D. Colocalization index between glutamylated tubulin and the indicated GFP-tagged proteins. Data represent the mean \pm S.E.M. n = 5, 3, 3, 4, and 4 cells from left to right. Two independent experiments.
- E. Normalized Linescan intensity profiles of GFP-CEP170C (green) and glutamylated tubulin (red) along the dashed line in (C).

References:

Rodríguez-Real G, Domínguez-Calvo A, Prados-Carvajal R, Bayona-Feliú A, Gomes-Pereira S, Balestra FR, Huertas P. Centriolar subdistal appendages promote double-strand break repair through homologous recombination. *EMBO Rep.* 2023 Oct 9;24(10):e56724.

Ma D, Wang F, Wang R, Hu Y, Chen Z, Huang N, Tian Y, Xia Y, Teng J, Chen J. α - γ -Taxilin are required for centriolar subdistal appendage assembly and microtubule organization. *Elife.* 2022 Feb 4;11:e73252.

3 - Figure 1E/F shows convincingly that CEP170C targets CCP5 deglutamylase to centrioles and this reduces by half the GT335 signal at centrosomes within 30 minutes. Can the authors quantify the signal at mother and daughter centrioles separately and test if there is a difference? This relates to the point that CEP170 localizes to SDA in mother centrioles and not daughter and if this is the same for CEP170C, there could be a difference in GT335 signal between the two (something that could be very interesting to test).

To evaluate whether there is a difference in the efficiency of glutamylation reduction between the mother and daughter centrioles, we measured the intensity of glutamylated tubulin at both centrioles. Our results showed no significant difference in glutamylated tubulin levels between the two centrioles before and after centrosomal hypoglutamylation, confirming that hypoglutamylation occurs uniformly at both centrioles (Fig EV1B).

Fig EV1B. Hypoglutamylation occurs uniformly at both centrioles in our system.

COS7 cells co-transfected with Ce3-FRB-CEP170C and the indicated constructs were treated with either 0.1% DMSO (green circles) or 100 nM rapamycin (Rapa, red circles) for 1 h, respectively. The treated cells were immunostained with the GT335 antibody, and the relative difference in glutamylated tubulin intensity between two centrioles were measured. Data represents mean \pm S.E.M. $n = 21, 25, 78, 18, 46,$ and 13 cells from left to right; 3 independent experiments. Statistical significance was determined using a one-way ANOVA test. “NS” indicates no significance difference among groups.

4- The authors demonstrate by various ways that centrioles integrity is not affected after 30 minutes of centrosomal deglutamylation (Figure 2). While this is convincing for the tested markers, it appears necessary to test longer depletion time as the glutamylation status might be important at certain stages of the centrosome function, for example during the G2/M transition when forces are applied to centrosomes. Also, the glutamylation status might be important during centriole assembly, something that cannot be tested with this short treatment. Can the authors test different longer timing?

As suggested by the reviewer, we have characterized the centrosome structure in cells expressing mCh-CEP170C, CCP5CD-mCh-CEP170C, or CCP5CDDM-mCh-CEP170C (Appendix Fig S6A-F). The centrioles, distal appendages, and pericentriolar matrix were labeled with Centrin2, CEP164, and Pericentrin, respectively. Expression of mCh-CEP170C, CCP5CD-mCh-CEP170C, or CCP5CDDM-mCh-CEP170C did not affect the pattern or intensity of Centrin2, CEP164, or Pericentrin, indicating that long-term hypoglutamylation at centrosomes has minimal impact on centrosome structure.

Appendix Figure S6. Long-term centrosomal hypoglutamylation suppresses microtubule nucleation and cilia formation but has minimal impact on centrosome structure.

A,C,E. COS7 cells transfected with the indicated constructs were immunostained with the antibodies against Centrin2 (A), CEP164 (C), and Pericentrin (E), respectively. Lower panels show magnified images of the areas demarcated by the dashed squares. Scale bar, 10 μ m.

B,D,F. The normalized intensity of Centrin2, CEP164, and Pericentrin of cells in (A), (C), and (E), respectively. The solid red lines and dashed red lines in violin plots represent the median and first/third quartiles, respectively. n (from left to right) = 121, 122, and 116 cells in (B); 307, 210, and 115 cells in (D); 194, 248, and 194 cells in (F); 3~6 independent experiments.

Students' *t*-tests were performed, and *P* values are indicated.

5- The title "centrosome glutamylation is dispensable for the structural integrity of centrosome" is not supported by the results. The authors only looked at a few markers, and if the PCM is reduced, this means that the centrosome integrity might be affected. If the authors were thinking about centrioles, the remaining 50% of the signal may be protecting the centriole, so this title/conclusion is too strong. Here the structural integrity is not fully analyzed, super-resolution microscopy more resolute than the SIM used, or a significant amount of electron microscopy imaging would be needed.

In this study, we used six markers to assess different components of centrosomes, including centrioles, subdistal appendages, distal appendages, and the pericentriolar matrix (**Fig 2**). Additionally, we employed structured illumination microscopy (SIM), a super-resolution imaging technique, to confirm that deglutamylation does not affect centriole length (**Fig 2K, M**). Characterizing centriole structure using electron microscopy is a time-intensive process. Given the constraints of the 3-month revision period, we have opted to refine our conclusion rather than conduct electron microscopy. Accordingly, we have revised the subtitle and conclusion to: "Centrosomal hypoglutamylation has minimal impact on centrosome structure."

6 -Concerning the "negative NEDD1" protein, can the authors do the co-sedimentation assay to show that it cannot interact with glutamylated tubulin? As these mutations are spread in the sequence, it is also possible that the non-centrosomal localization is due to protein misfolding and not to a lack of electrostatic interactions. This point needs to be tested, and numerical simulation does not appear to be sufficient. NEDD1 is anchored to the centrosome by CEP192 and depletion of CEP192 leads to loss of NEDD1 at the centrosome (<https://doi.org/10.1038/s41467-021-26252-5>). A rescue with CEP192 relocated to the centrosome would be interesting to determine whether NEDD1 relocates and whether nucleation is restored.

To minimize undeserved impact on protein structure, positively charged residues on the surface and in regions lacking predicted secondary structures of NEDD1 were replaced with negatively charged residues. We also carefully evaluated whether the introduced mutations in negative NEDD1 could cause protein misfolding using AlphaFold, a leading computational method for predicting 3D protein structures (Jumper et al., *Nature*, 2021). The results indicated that negative NEDD1 folds normally and maintains a structure highly similar to wild-type NEDD1 (Appendix Fig S7).

As suggested by the reviewer, we evaluated whether CEP192 is sufficient to recruit NEDD1 to hypoglutamylated centrosomes. To anchor CEP192 at the centrosome, we tagged it with CEP170C, generating the construct CEP192-Ce3-FRB-CEP170C. This construct successfully restored NEDD1 and γ -tubulin to normal levels at hypoglutamylated centrosomes (Fig EV2). These results suggest that CEP192 also contribute to NEDD1/ γ -tubulin recruitment.

Figure EV2. CEP192 is sufficient to recruit NEDD1 and γ -tubulin to hypoglutamylated centrosomes.

- A,C. COS7 cells co-transfected with the indicated constructs were treated with DMSO (0.1%) or Rapa (100 nM rapamycin) for 1 h. Following treatment, cells were immunostained with antibodies against NEDD1 (A) and γ -tubulin (C), respectively. Scale bar, 10 μ m.
- B,D. Quantification of the normalized intensity of NEDD1 (B) in cells from (A) and γ -tubulin (D) in cells from (C). Data represent as mean \pm S.E.M. n (from left to right) = 62, 51, 84, 101, 67, 55, 53, and 54 cells in (B); 143, 152, 134, 135, 37, 36, 85, and 69 cells in (D); 3~6 independent experiments. Students' *t*-tests were performed, and *P* values are indicated.

Reference:

Tunyasuvunakool K, Adler J, Wu Z, Green T, Zielinski M, Židek A, Bridgland A, Cowie A, Meyer C, Laydon A, Velankar S, Kleywegt GJ, Bateman A, Evans R, Pritzel A, Figurnov M, Ronneberger O, Bates R, Kohl SAA, Potapenko A, Ballard AJ, Romera-Paredes B, Nikolov S, Jain R, Clancy E, Reiman D, Petersen S, Senior AW, Kavukcuoglu K, Birney E, Kohli P, Jumper J, Hassabis D. Highly accurate protein structure prediction for the human proteome. *Nature*. 2021 Aug;596(7873):590-596.

7- Upon deglutamylation (Figure 6C), the cilium length is only slightly affected. Could the authors test after a longer SS? For example, what about after 24h? In addition, the authors are using an acute depletion here, so glutamylation may return to its normal level. Could the authors quantify the GT335 signal after 0h, 4h, and 24h?

It is also surprising to see in graph 6C bars at the 0-micron level, suggesting that unformed cilia are included in the quantification of cilia length, is it correct? Could the authors exclude those and redo the statistics as this can impact the results?

As suggested by the reviewer, we measured the length of cilia extending from hypoglutamylated centrosomes after 24 h of serum starvation. Our results indicated that centrosomal hypoglutamylation significantly reduced both cilia length and the proportion of ciliated cells compared to two control groups after 24 h of serum starvation (Fig 6A-D).

Rapamycin-inducible dimerization is widely considered an irreversible process due to the extremely high affinity between rapamycin and FKBP, resulting in slow clearance from cells (~30 h) (Lin et al., *Angew Chem Int Ed Engl*, 2013; Voß et al., *Curr Opin Chem Biol*, 2015; Putyrski and Schultz, *FEBS Lett.*, 2012). This property enables a sustained reaction for centrosomal hypoglutamylation. Indeed, even 24 h after rapamycin-induced recruitment of CCP5CD to centrosomes, centrosomal hypoglutamylation was still observed compared to the control groups (Fig EV1C).

For the evaluation of cilia length, non-ciliated cells were excluded from the analysis, and statistical analysis was performed (Fig. 6C and Appendix Fig S6J).

Figure 6. Centrosomal glutamylation is required for ciliogenesis and cilia maintenance

A. Experimental protocol to induce acute centrosome hypoglutamylation followed by serum starvation–induced ciliogenesis. Cells were serum-starved for 4 h or 24 h, after which nascent and mature primary cilia were stained for immunofluorescence analysis (IF).

B. NIH3T3 cells co-transfected with Ce3-FRB-CEP170C and either CCP5CD-Neon-FKBP or CCP5CDDM-Neon-FKBP were serum-starved (SS) for the indicated times according to the

protocol in (A). Cilia were labeled by anti-Arl13B antibody. The right panels show magnified images of the areas outlined by dashed squares. Scale bar, 10 μ m.

C. Quantification of cilium length in cells from (B). Data (black) represent the mean \pm S.E.M. $n = 225$, 464, and 392 cells in Neon-FKBP, CCP5CD-Neon-FKBP, and CCP5CDDM-Neon-FKBP groups, respectively, obtained from 8~10 independent experiments.

D. Percentage of ciliated cells in (B). Data represent the mean \pm S.E.M. $n = 534$, 399, and 446 cells in Neon-FKBP, CCP5CD-Neon-FKBP, and CCP5CDDM-Neon-FKBP groups, respectively, obtained from 3 independent experiments.

Student's *t*-tests were performed, and *P* values are indicated.

Fig EV1C. Long-term hypoglutamylatation at centrosomes.

COS7 cells co-transfected with Ce3-FRB-CEP170C and the indicated constructs were treated with either DMSO (0.1%; green circles) or Rapa (100 nM rapamycin; red circles) for 24 h. Treated cells were immunostained with GT335 antibody. The normalized intensity of glutamylated tubulin at centrosome are shown as mean \pm S.E.M. $n = 47$, 27, 71, 46, 85, and 22 cells from left to right; 3 independent experiments. Student's *t*-tests were performed, and *P* values are indicated.

References:

Yu-Chun Lin, Yuta Nihongaki, Tzu-Yu Liu, Shiva Razavi, Moritoshi Sato, Takanari Inoue. Rapidly reversible manipulation of molecular activity with dual chemical dimerizers. *Angew Chem Int Ed Engl.* 2013, 52(25):6450-4.

Stephanie Voß, Laura Klewer, Yao-Wen Wu. Chemically induced dimerization: reversible and spatiotemporal control of protein function in cells. *Curr Opin Chem Biol.* 2015, 28:194-201.

Mateusz Putyrski, Carsten Schultz. Protein translocation as a tool: The current rapamycin story. *FEBS Lett.* 2012, 586(15):2097-105.

8- Although CCP5 is recruited to the centrosome, the enzyme is still active and also resides in the

cytosol, probably affecting cytoplasmic microtubules. Did the authors look at the level of glutamylation in the cytosol? The effect on the satellites may come from a trafficking problem on the microtubules and not directly at the centrosome. This effect can be similar for the ciliogenesis. In general, the authors should test the level of glutamylation (GT335) more systematically in ciliogenesis.

As suggested by the reviewer, we measured the level of glutamylated tubulin in the cytosol before and after centrosomal hypoglutamylation. Consistent with our previous study (Hong *et al.*, Nat Commun., 2018), the expression of CCP5CD did not alter tubulin glutamylation levels in the cytosol compared to the two control groups. Furthermore, the recruitment of CCP5CD to centrosomes did not affect cytoplasmic glutamylated tubulin. These results confirmed the specific reduction of glutamylated tubulin at centrosomes using our system (Fig EV1A).

Fig EV1A. Centrosomal hypoglutamylation does not affect tubulin glutamylation in cytosol.

COS7 cells co-transfected with Ce3-FRB-CEP170C and the indicated constructs treated with either 0.1% DMSO (green circles) or 100 nM rapamycin (Rapa, red circles) for 1 h. Treated cells were immunostained with GT335 antibody and the level of glutamylated tubulin in cytosol was measured. Data represent the mean \pm S.E.M. $n = 23, 18, 22, 39,$ and 23 cells from left to right; 3 independent experiments. Statistical analysis was performed using a one-way ANOVA test. "NS" indicates no significance difference among groups.

Reference:

Hong SR, Wang CL, Huang YS, Chang YC, Chang YC, Pusapati GV, Lin CY, Hsu N, Cheng HC, Chiang YC, Huang WE, Shaner NC, Rohatgi R, Inoue T, Lin YC. Spatiotemporal manipulation of ciliary glutamylation reveals its roles in intraciliary trafficking and Hedgehog signaling. Nat Commun. 2018 Apr 30;9(1):1732.

9-The experiment in Figure S7 is very important but indicates that the phenotype on ciliogenesis is minimal. What about the number of cilia? the level of cytoplasmic glutamylation?

Other than ciliogenesis, what about a long-term depletion and the effects on the distribution of EB1, the formation of spindles in mitosis?

We calculated the percentage of ciliated cells expressing mCh-CEP170C, CCP5CD-mCh-CEP170C, and CCP5CDDM-mCh-CEP170C, respectively. The expression of CCP5CD-mCh-CEP170C significantly decreased the cilia length and reduced the proportion of ciliated cells compared to the two control groups (Appendix Fig S6I-K), demonstrating that long-term hypoglutamylation at centrosomes suppresses cilia formation.

We also confirmed that centrosomal hypoglutamylation induced by our system does not affect tubulin glutamylation in the cytosol (Fig EV1A)

As suggested by the reviewer, we evaluated the effect of long-term centrosomal hypoglutamylation on EB1 pattern. By tracking EB1 trajectories, we found that the expression of CCP5CD-mCh-CEP170C but not mCh-CEP170C or CCP5CDDM-mCh-CEP170C, significantly reduced the number of EB1 tracks originating from centrosomes (Appendix Fig S6G and H). This result confirms that long-term centrosomal hypoglutamylation suppresses microtubule nucleation.

We also assessed the effects of centrosomal hypoglutamylation on mitotic spindle formation. HeLa cells were synchronized in G2/M and released into mitosis immediately after centrosomal hypoglutamylation treatment (Fig 8). Most control cells exhibited bipolar mitotic spindles with properly aligned chromosomes at the metaphase plate. However, centrosomal hypoglutamylation resulted in aberrant mitotic spindles and altered chromosome distribution (Fig 8B and C). Moreover, these defects in mitotic spindle formation and chromosome alignment significantly prolonged the mitosis (Fig 8D and E). These results demonstrated that centrosomal glutamylation is crucial for mitotic spindle formation and proper cell division.

Appendix Figure S6. Long-term hypoglutamylation at centrosomes suppresses cilia formation.

- I. NIH3T3 cells transfected with the indicated constructs were serum-starved for 24 h and then immunostained with Arl13 antibody (green) to visualize primary cilia. Scale bar, 10 μm .
- J,K. Quantification of cilia length (J) and proportion of ciliated cells (K) in the cells shown in (I). Data are presented as mean \pm S.E.M. n (from left to right) = 72, 82, and 114 cells in (J); from 3-6 independent experiments. Students' *t*-tests were performed, and *P* values are indicated.

Fig EV1A. Centrosomal hypoglutamylation does not affect tubulin glutamylation in cytosol.

COS7 cells co-transfected with Ce3-FRB-CEP170C and the indicated constructs treated with either 0.1% DMSO (green circles) or 100 nM rapamycin (Rapa, red circles) for 1 h. Treated cells were immunostained with GT335 antibody and the level of glutamylated tubulin in cytosol was measured. Data represent the mean \pm S.E.M. $n = 23, 18, 22, 39,$ and 23 cells from left to right; 3 independent experiments. Statistical analysis was performed using a one-way ANOVA test. "NS" indicates no significance difference among groups.

Appendix Fig S6G, H . Long-term centrosomal hypoglutamylation suppresses centrosome-derived microtubules.

G. COS7 cells were co-transfected with EB1-YFP and the indicated constructs. EB1-YFP comet trajectories were visualized using live-cell imaging. Maximum projections of EB1-YFP from 2.5-min imaging are shown. Arrows indicate centrosome regions. Scale bar, $10 \mu\text{m}$.

H. The number of EB1-YFP tracks emitted from centrosomes in cells from (G) is shown. The solid red lines and dashed red lines in violin plots represent the median and first/third quartiles, respectively. $n = 48, 68, 56$ cells from left to right; 3 independent experiments.

Figure 8. Centrosomal hypoglutamylation perturbs mitotic spindle formation and prolongs mitosis

- A. Schematic of the protocol for RO-3306-induced cell synchronization at the G2/M transition, followed by rapamycin (Rapa)-induced centrosomal hypoglutamylation after RO-3306 washout (W/O). Cell morphology was analyzed by immunofluorescence (IF) 30 min after rapamycin or DMSO treatment.
- B. HeLa cells co-transfected with the indicated constructs were synchronized at the G2/M transition and released into mitosis for 30 min in the presence of DMSO (0.1%) or rapamycin (Rapa, 100 nM). Mitotic spindles and chromosomes were labeled with anti- α -tubulin antibody (green) and DAPI (blue), respectively. Dashed lines indicate cell boundaries. Scale bar, 10 μ m.
- C. Quantification of the percentage of cells displaying a normal mitotic spindle pattern as shown in (B). Data are presented as mean \pm S.E.M. $n = 90$ and 78 cells in CCP5CD and CCP5CDDM groups, respectively; 3 independent experiments.

D. Representative video frames of HeLa cells co-transfected with the indicated constructs, captured 30 min after treatment with DMSO (0.1%) or rapamycin (100 nM). Red dashed lines indicate the boundaries of mitotic cells. Scale bar, 10 μ m.

E. Quantification of mitosis duration (from metaphase to telophase) in HeLa cells co-transfected with mCh-FRB-CEP170C and the indicated constructs, treated with DMSO (0.1%) or rapamycin (100 nM). Data are presented as mean \pm SD (black). n = 90 and 78 cells in the CCP5CD and CCP5CDDM groups, respectively. Three independent experiments.

Student's *t*-tests were performed, and *P* values are indicated.

10 -Overall, the conclusions should be tone down, as the absence of glutamylation is not proven, it seems to rather be an hypo-glutamylation, so the headings in the sub-sections and the discussion should be adapted.

As suggested by the reviewer, we toned down the conclusion, particularly by replacing "deglutamylation" with "hypoglutamylation".

- minor concerns that should be addressed

-In the abstract, it is written r-tubulin and not gamma tubulin.

We have corrected it accordingly.

- For the EB1-YFP experiments, the authors mention the "EB1 dynamics", here the dynamics are not tested, only the number of comets.

We have corrected it accordingly.

- as a comment: it has been shown by cryo-tomography that IFT trains interact above the basal body (transition zone: DOI: 10.1126/science.abm6704), so the result obtained by the authors about the absence of phenotype on the IFT is logical.

We thank the reviewer for the valuable information and added the reference (van den Hoek et al., Science, 2022) to the revised manuscript.

Reference:

van den Hoek H, Klena N, Jordan MA, Alvarez Viar G, Righetto RD, Schaffer M, Erdmann PS, Wan W, Geimer S, Plitzko JM, Baumeister W, Pigino G, Hamel V, Guichard P, Engel BD. In situ architecture of the ciliary base reveals the stepwise assembly of intraflagellar transport trains. Science. 2022 Jul 29;377(6605):543-548.

-The Figure 2K with the 3D SIM figure shows that CCP5CD is located around the centrioles, more like PCM than at the centrioles themselves. The resolution makes it difficult to see the microtubule wall.

We thank the reviewer for bringing this to our attention. CCP5CD was recruited to CEP170C-labeled area, which is partially localized to both the centriole and PCM (Appendix Fig S2B and C). That explains why CCP5CD is observed at and around the centrioles in Fig 2K.

- The impact of NEDD1 after centrosome deglutamylation affects MT nucleation, which is probably the cause of defects in satellite trafficking and the resulting defects in ciliogenesis. Perhaps the authors could better discuss this point.

We have described how NEDD1 anchors to glutamylated centrosomes and directly regulates microtubule nucleation while indirectly influencing satellite trafficking and ciliogenesis in the revised manuscript.

Dr. Yu-Chun (Frank) Lin
National Tsing Hua University
Institute of Molecular Medicine, College of Life Science
101, Section 2, Kuang-Fu Road
Hsinchu City 300044
Taiwan

26th Mar 2025

Re: EMBOJ-2024-118291R
Centrosomal Glutamylation Recruits the Microtubule Nucleation Factors to Ensure Its Functions

Dear Dr. Lin,

Thank you for submitting your revised manuscript for our consideration. Two of the original referees have now assessed it once more, and were fully satisfied with the revisions, except for some textual amendments asked by referee 3, which I would ask you to incorporate in a final round of minor revision.

At this point, please also address the remaining editorial issues, as follows:

- Most importantly, we still need you to complete and upload the Source Data Checklist that had been sent to you by our Source Data curator, Hannah Sonntag (I am attaching it once more to this message). Please also double-check to make sure that all requested Source Data items have been uploaded.
- The format of the reference list (journal abbreviations, complete volume/page information) needs to be consistently adjusted to EMBO Journal style (as specified in our Guide to Authors), removing any hyperlinks, DOIs or inappropriate "preprint" labels.
- As we are switching from a free-text author contribution statement towards a more formal statement based on Contributor Role Taxonomy (CRediT) terms, please remove the present Author Contribution section and instead specify each author's contribution(s) directly in the Author Information page of our submission system during upload of the final manuscript. See <https://casrai.org/credit/> for more information.
- Please provide suggestions for a short 'blurb' text prefacing and summing up the study in two sentences (max. 250 characters), followed by 3-5 one-sentence 'bullet points' with brief factual statements of key results of the paper; they will form the basis of an editor-written 'Synopsis' accompanying the online version of the article. Please also upload a synopsis image, which can be used as a "visual title" for the synopsis section of your paper. The image should be in PNG or JPG format with the modest dimensions of EXACTLY 550 pixels wide and 300-600 pixels high.
- Finally, we note that the legends for figures 2, 4, EV2 are still not provided in a sequential manner (i.e., A-B-C...) as previously requested. My suggestion for solving this would be to alter the labeling within the figure file (e.g., Fig 2A-C-E becoming Fig 2A-B-C, and Fig 2B-D-F becoming Fig 2 D-E-F, etc.), which would allow you to retain the non-duplicated format of the figure legends?

I am therefore returning the manuscript to you for a final round of revision, to allow you to make these modifications and upload the revised files. Once we will have received them, we should be ready to swiftly proceed with formal acceptance and production of the manuscript.

Yours sincerely,

Hartmut Vodermaier

- 1) Every manuscript requires a Data Availability section (even if only stating that no deposited datasets are included). Primary datasets or computer code produced in the current study have to be deposited in appropriate public repositories prior to resubmission, and reviewer access details provided in case that public access is not yet allowed. Further information: embopress.org/page/journal/14602075/authorguide#dataavailability
- 2) Each figure legend must specify
 - size of the scale bars that are mandatory for all micrograph panels
 - the statistical test used to generate error bars and P-values
 - the type error bars (e.g., S.E.M., S.D.)
 - the number (n) and nature (biological or technical replicate) of independent experiments underlying each data point
 - Figures may not include error bars for experiments with $n < 3$; scatter plots showing individual data points should be used instead.
- 3) Revised manuscript text (including main tables, and figure legends for main and EV figures) has to be submitted as editable text file (e.g., .docx format). We encourage highlighting of changes (e.g., via text color) for the referees' reference.
- 4) Each main and each Expanded View (EV) figure should be uploaded as individual production-quality files (preferably in .eps, .tif, .jpg formats). For suggestions on figure preparation/layout, please refer to our Figure Preparation Guidelines: <http://bit.ly/EMBOPressFigurePreparationGuideline>
- 5) Point-by-point response letters should include the original referee comments in full together with your detailed responses to them (and to specific editor requests if applicable), and also be uploaded as editable (e.g., .docx) text files.
- 6) Please complete our Author Checklist, and make sure that information entered into the checklist is also reflected in the manuscript; the checklist will be available to readers as part of the Review Process File. A download link is found at the top of our Guide to Authors: embopress.org/page/journal/14602075/authorguide
- 7) All authors listed as (co-)corresponding need to deposit, in their respective author profiles in our submission system, a unique ORCID identifier linked to their name. Please see our Guide to Authors for detailed instructions.
- 8) Please note that supplementary information at EMBO Press has been superseded by the 'Expanded View' for inclusion of additional figures, tables, movies or datasets; with up to five EV Figures being typeset and directly accessible in the HTML version of the article. For details and guidance, please refer to: embopress.org/page/journal/14602075/authorguide#expandedview
- 9) To facilitate reproducibility and cross-laboratory adoption of methodologies, please structure the Materials & Methods section as outlined in our guide to authors, including a completed Reagents and Tools Table that can be downloaded from our author guidelines as well (<https://www.embopress.org/page/journal/14602075/authorguide#structuredmethods>).
- 10) Digital image enhancement is acceptable practice, as long as it accurately represents the original data and conforms to community standards. If a figure has been subjected to significant electronic manipulation, this must be clearly noted in the figure legend and/or the 'Materials and Methods' section. The editors reserve the right to request original versions of figures and the original images that were used to assemble the figure. Finally, we generally encourage uploading of numerical as well as gel/blot image source data; for details see: embopress.org/page/journal/14602075/authorguide#sourcedata

At EMBO Press, we ask authors to provide source data for the main manuscript figures. Our source data coordinator will contact you to discuss which figure panels we would need source data for and will also provide you with helpful tips on how to upload and organize the files.

In the interest of ensuring the conceptual advance provided by the work, we recommend submitting a revision within 3 months (24th Jun 2025). Please discuss the revision progress ahead of this time with the editor if you require more time to complete the revisions. Use the link below to submit your revision:

Link Not Available

Referee #2:

The authors have sufficiently addressed all the queries, I have no further comments. The manuscript is suitable for publication now.

Referee #3:

Overall, the authors responded satisfactorily to my requests, and I recommend publication of the manuscript.

I have however, one comment left listed below:

In response to Reviewer 2, the authors demonstrate that Polyglutamylation is not affected at the centrosomes, and that only glutamylation is influenced by CCP5. Given that the negatively charged protein NEDD1 is recruited to centrosomes due to the positive charges introduced by both glutamylation and polyglutamylation (as suggested in Guichard et al, 2023: <https://doi.org/10.1016/j.semcdb.2021.12.001>, which the authors have overlooked, please include this citation in the next revision), it would be helpful if the authors could include a sentence in the discussion how NEDD1 might be impacted by the sole reduction of glutamylation.

We would like to thank the reviewers and the editors for their constructive feedback and insightful suggestions. We have revised the manuscript accordingly and have provided a separate file with tracked changes.

Referee #2:

The authors have sufficiently addressed all the queries, I have no further comments. The manuscript is suitable for publication now.

We appreciate the reviewer's expert comments and support.

Referee #3:

Overall, the authors responded satisfactorily to my requests, and I recommend publication of the manuscript.

I have however, one comment left listed below:

In response to Reviewer 2, the authors demonstrate that Polyglutamylation is not affected at the centrosomes, and that only glutamylation is influenced by CCP5. Given that the negatively charged protein NEDD1 is recruited to centrosomes due to the positive charges introduced by both glutamylation and polyglutamylation (as suggested in Guichard et al, 2023: <https://doi.org/10.1016/j.semcdb.2021.12.001>, which the authors have overlooked, please include this citation in the next revision), it would be helpful if the authors could include a sentence in the discussion how NEDD1 might be impacted by the sole reduction of glutamylation.

We thank the reviewer for the constructive suggestion and have included the citation in the revised manuscript.

Dr. Yu-Chun (Frank) Lin
National Tsing Hua University
Institute of Molecular Medicine, College of Life Science
101, Section 2, Kuang-Fu Road
Hsinchu City 300044
Taiwan

28th Mar 2025

Re: EMBOJ-2024-118291R1
Glutamylation of Centrosomes Ensures their Function by Recruiting Microtubule Nucleation Factors

Dear Dr. Lin,

Thank you for submitting your final revised manuscript for our consideration. I am pleased to inform you that we have now accepted it for publication in The EMBO Journal.

Yours sincerely,

Hartmut Vodermaier
